# OPERATOR LEARNING FOR FAMILIES OF FINITE-STATE MEAN-FIELD GAMES

## ABSTRACT

Finite-state mean-field games (MFGs) arise as limits of large interacting particle systems and are governed by an MFG system, a coupled forward–backward differential equation consisting of a forward Kolmogorov–Fokker–Planck (KFP) equation describing the population distribution and a backward Hamilton–Jacobi–Bellman (HJB) equation defining the value function. Solving MFG systems efficiently is challenging, with the structure of each system depending on an initial distribution of players and the terminal cost of the game. We propose an operator learning framework that solves parametric families of MFGs, enabling generalization without retraining for new initial distributions and terminal costs. We provide theoretical guarantees on the approximation error, parametric complexity, and generalization performance of our method, based on a novel regularity result for an appropriately defined flow map corresponding to an MFG system. We then demonstrate empirically that our framework achieves accurate approximation for two representative instances of MFGs: a cybersecurity example and a high-dimensional quadratic model commonly used as a benchmark for numerical methods for MFGs.

## 1 INTRODUCTION

Mean-field games (MFGs), introduced by Huang et al. (2006) and Lasry & Lions (2007), model the behavior of stochastic games with many identical players by considering the limiting situation with an infinite population. While a large portion of the corresponding literature considers continuous state spaces, MFGs with finite state spaces find applications in economics, epidemic prevention, cybersecurity, resource allocation, and multi-agent reinforcement learning, and beyond (Gomes et al., 2014; Kolokoltsov & Bensoussan, 2016; Aurell et al., 2022; Mao et al., 2022; Yardim & He, 2025). The theory of MFGs is well-established, with results concerning existence, uniqueness, and connections with finite-player games in Gomes et al. (2013); Bayraktar & Cohen (2018); Cecchin & Pelino (2019); see the books Carmona & Delarue (2018a;b) for more background. Nonetheless, numerically solving finite-state MFGs remains challenging, especially over large state spaces.

Machine learning-based methods have proven promising for overcoming the numerical challenges associated high-dimensional MFGs, in both continuous and finite state spaces; see Fouque & Zhang (2020); Carmona & Laurière (2021; 2022); Min & Hu (2021); Han et al. (2024) for deep learning methods and Guo et al. (2019); Subramanian & Mahajan (2019); Elie et al. (2020); Cui & Koeppl (2021) for reinforcement learning methods. However, these methods treat each MFG individually, requiring the user to rerun the method anew for each MFG instance. Several recent works, such as Cohen et al. (2024), propose more general methods to learn MFGs equilibria as a function of the initial distribution by exploiting the connection with the master equation, a nonlinear PDE characterizing finite-state MFGs (Cardaliaguet et al., 2019). However, these methods rely on problem-specific loss functions and cannot be extended to learn MFG equilibria as a function of the model's parameters such as its cost functions.

In this work, we frame MFG equilibria as outputs of an *operator*, called the **flow map**, which maps initial distributions and cost functions to the corresponding Nash equilibrium. We then train a neural network (NN) to learn this operator.

**Main Contributions.** Our main contributions are as follows:

- **Algorithm:** We combine Picard iteration and operator learning to approximate the flow map operator for parametrized families of finite-state MFGs (see Fig. 1).
- **Parametric complexity:** We prove that the flow map can be approximated to accuracy $\mathcal{O}(K^{-1/(d+k+2)})$ using an NN with width $W = \mathcal{O}(K^{(2(d+k)+3)/(2(d+k)+4)})$ and depth $L = \mathcal{O}(\log(d+k+1))$, where $d$ is the number of states, $K$ is a bound on the NN weights, and $k$ is the dimension of the set of parameters specifying the family of MFGs.
- **Generalization error:** We prove that for such $W$ and $L$, given $n$ samples produced via Picard iteration, our method's generalization error is bounded by $\mathcal{O}(n^{-1/(d+k+4)}\log(n))$.
- **Numerical experiments:** We demonstrate the accuracy and scalability of our method on two standard finite-state MFG benchmarks.

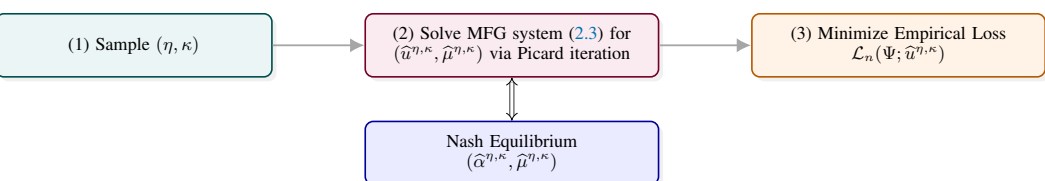

Figure 1: Given (1) sample initial distributions $\eta$ and cost parameters $\kappa$, we bypass the need to compute the optimal controls and flow of measures (Nash equilibrium) of an MFG by (2) solving the MFG system via Picard iteration. We then use the resulting trajectories to (3) approximate the solution operator for the family using a neural network, trained by minimizing an empirical loss over the samples from (1). In practice, the last step uses stochastic gradient descent (see Algo. 1).

**Operator Learning.** Independent from the literature on MFGs is that of *operator learning*, an umbrella term that typically describes machine learning methods for approximating maps between function spaces. One natural application of such methods is to partial differential equations (PDEs), and the general framework has been applied with impressive success to fluid dynamics in Li et al. (2021); Kovachki et al. (2023), astrophysics in Mao et al. (2023), and large-scale weather forecasting in Kurth et al. (2023); Lam et al. (2023). In most applications, one attempts to learn the operator that maps the initial data of a PDE, belonging to some Banach space, to its solution, belonging to a potentially distinct Banach space; see Kovachki et al. (2024); Boullé & Townsend (2024), for overviews of the field of operator learning from a mathematical perspective. The development of novel architectures for operator learning, as in Li et al. (2021) and Lu et al. (2021); Wang et al. (2021) has allowed for its recent empirical success.

However, instead of leveraging architectural advances in the field, our insight is inspired by the work of Lanthaler & Stuart (2025). The authors introduce the *curse of parametric complexity* for operator learning: given any compact subset $K$ of an infinite-dimensional Banach space, there exists an operator from $K$ into another Banach space that can only be approximated with a functional of neural network type (i.e., the composition of a linear operator to a Euclidean space and a neural network) whose width and depth are exponential in the approximation error. Lanthaler & Stuart (2025) circumvent this issue for first-order Hamilton–Jacobi–Bellman (HJB) equations with an initial condition, learning the operator that maps initial conditions to solutions. Associated with each HJB equation is a system of ODEs, also referred to as the characteristics of the PDE. By learning the flow map for the characteristics and then reconstructing the solution by interpolation, (Lanthaler & Stuart, 2025, Theorem 5.1) beats the so-called curse of parametric complexity, enabling operator learning with neural networks of bounded width and depth using a method they label HJ-Nets. Given the similarity between the forward-backward ODE system for MFGs and the characteristics of first-order HJB equations, we take this as inspiration for our approach to learning MFG equilibria; see Appendix F for a more in-depth comparison.

**Related Work.** We clarify the connection between our contributions and several closely related works on MFGs. Cohen et al. (2024) proposes and analyzes two methods to solve the master equation for finite-state MFGs, handling varying initial distributions. However, their methods do not generalize to the setting of MFGs with varying cost functions as we consider. Chen et al. (2023); Huang & Lai (2025) proposes operator learning methods for continuous space and time MFGs by learning the solution as a function of the initial distribution. Although philosophically similar to our operator learning approach, their methods do not apply to finite-state space MFGs, and neither method provides a solution for parametrized families of MFGs with varying cost functions. Fi-

nally, reinforcement learning methods for population-dependent policies tackle discrete time MFGs (see Perrin et al. (2022); Li et al. (2023); Zhang et al. (2025); Wu et al. (2025) for recent work in this domain), while we focus on continuous time models. To our knowledge, even in discrete time, no method has been proposed to solve parameterized families of MFGs at once.

**Organization.** In Section 2, we describe finite-state MFGs and the forward-backward ODE system that characterizes MFG equilibria, including the assumptions that we place on parametrized families of MFGs. Next, we describe the flow map, mapping parameters to equilibria. In Section 3, we describe our operator learning method in detail. In Section 4, we present the associated approximation, parametric complexity, and generalization guarantees, with technical proofs in the appendix. Finally, in Section 5, we provide numerical experiments for two finite-state MFGs often used as benchmarks for numerical methods: a simple model of cybersecurity and a high-dimensional quadratic model.

## 2 BACKGROUND

We first provide provide a comprehensive overview of finite-state MFGs for the unfamiliar reader in Section 2.1, referring to Appendix B for more details. Then, in Section 2.2, we describe the object that we seek to approximate via operator learning: the flow map for a parametrized family of MFGs.

### 2.1 FINITE-STATE MFGS

**Controls and state dynamics.** In a finite-state MFG, a representative player chooses *Markovian controls* taking values in a compact set of rates, $\mathbb{A} \subseteq \mathbb{R}_+ := [0, \infty)$. Specifically, the player's control $\alpha$ is a time-dependent $d \times d$ matrix with values in $\mathbb{A}$, with rows $(\alpha_y(t, x))_{y \in [d]}$ and individual entries $\alpha_y(t, x)$ determining the rate of transition between state $x$ to state $y$ at time $t$. When starting with initial distribution $\eta$ and using control $\alpha$, the player's state, denoted by $\mathcal{X}_t^{\eta, \alpha} \in [d]$ at time $t$, obeys the dynamics of a continuous-time Markov chain with $X_0^{\eta, \alpha} \sim \eta$ and:

$$\Pr(\mathcal{X}_{t+h}^{\eta, \alpha} = y \mid \mathcal{X}_t^{\eta, \alpha} = x) = \alpha_y(t, x)h + o(h), \quad h \to 0^+. \tag{2.1}$$

**Cost function.** The representative player aims to minimize a cost functional over the time interval $[0, T]$. The cost depends not only on the player's chosen control and state at time $t \in [0, T]$, but also on the population distribution $\mu(t) \in \mathcal{P}([d])$, where $\mathcal{P}([d])$ is the set of probability measures on $[d] := \{1, \dots, d\}$, identifiable with the probability simplex in $\mathbb{R}^d$. We denote by $g$ the terminal cost and $f, F$ two running costs depending on the player's chosen control and the population distribution, respectively. If the population distribution's flow $\mu = (\mu(s))_{s \in [0, T]}$ is given, the representative player aims to minimize the total expected cost over controls $\alpha = (\alpha_y(s, x))_{s \in [0, T], x \in [d], y \in [d]}$ :

$$J_\eta(\alpha, \mu) = \mathbb{E}\left[\int_0^T \Big(f(\mathcal{X}_s^{\eta, \alpha}, \alpha(s, \mathcal{X}_s^{\eta, \alpha})) + F(\mathcal{X}_s^{\eta, \alpha}, \mu(s))\Big)ds + g(\mathcal{X}_T^{\eta, \alpha}, \mu(T))\right]. \tag{2.2}$$

Notice that since $\mu$ is a deterministic flow of measures and $\mu(0) = \eta$ is fixed, the control may depend implicitly on the population distribution through time. When $\mu$ is given, this is a standard stochastic optimal control problem. However, $\mu$ should be determined endogenously as the population evolution resulting from the players' optimal behavior.

**MFG equilibrium.** This leads us naturally to the idea of an MFG equilibrium, a form of Nash equilibrium in which the population distribution is the same as the representative player's distribution.

**Definition 2.1.** *An **MFG equilibrium** for an initial distribution $\eta \in \mathcal{P}([d])$ is a pair $(\overline{\alpha}, \overline{\mu})$ such that: (1) $\overline{\alpha}$ minimizes the cost functional $J_\eta(\cdot, \overline{\mu})$ and (2) for every $t \in [0, T]$, $\overline{\mu}(t) = \mathrm{Law}(\mathcal{X}_t^{\eta, \overline{\alpha}})$.*

Observe that the MFG equilibrium *depends* on the initial distribution $\eta \in \mathcal{P}([d])$. This presents one of the primary difficulties that we aim to address: can one efficiently compute MFG equilibria simultaneously *for arbitrary initial distributions*? Before tackling this question, we first explain how one can solve an MFG for a *fixed* initial distribution.

**Forward-backward ODE system.** The two points in Definition 2.1 can be translated into two equations: one for the value function $u(t, x)$ of the representative player (i.e., the optimal cost attainable at time $t$ in state $x$), and one for the evolution of the population distribution. In finite-state, continuous-time MFGs, both take the form of ordinary differential equations (ODEs). Then,

MFG equilibria can be characterized as solutions of a forward–backward system of coupled ODEs, each in dimension $d$. More precisely, $(\overline{\alpha}, \overline{\mu})$ is an MFG equilibrium if and only if $\overline{\alpha}_y(t, x) = \gamma_x^*(y, \Delta_y u(t, \cdot)) := \arg\min_a \{f(x, a) + a \cdot \Delta_y u(t, \cdot)\}$ where $\Delta_x f := (f(y) - f(x))_{y \in [d]} \in \mathbb{R}^d$ plays the role of a discrete gradient and $(u, \mu)$ solve the **MFG system**:

$$\begin{cases} \frac{d}{dt} u(t, x) + \bar{H}(x, \mu(t), \Delta_x u(t, \cdot)) = 0, & (t, x) \in [0, T] \times [d] \quad \textbf{(HJB)} \\ \frac{d}{dt} \mu(t, x) = \sum_{y \in [d]} \mu(t, y) \gamma_x^*(y, \Delta_y u(t, \cdot)), & (t, x) \in [0, T] \times [d] \quad \textbf{(KFP)} \\ \mu(0, x) = \eta(x), \quad u(T, x) = g(x, \mu(T)), & x \in [d], \end{cases} \quad (2.3)$$

with $\bar{H}$ being the extended Hamiltonian of the representative player's control problem, defined in terms of the **Hamiltonian** $H$ as:

$$H(x, p) := \min_a \left\{ f(x, a) + \sum_{y \neq x} a_y p_y \right\}, \qquad \bar{H}(x, \eta, p) := H(x, p) + F(x, \eta). \quad (2.4)$$

We will sometimes write $u^\eta$ and $\mu^\eta$ to stress the dependence on the initial distribution $\eta$. We refer to the first equation as **Hamilton–Jacobi–Bellman (HJB)** equation and to the second equation as the **Kolmogorov–Fokker–Planck (KFP)** equation.

The above MFG system admits a unique solution under standard assumptions; see Appendix B and Bayraktar & Cohen (2018); Cecchin & Pelino (2019) for more details. For simplicity, we focus on the following sufficient condition:

**Assumption 2.1.** *The minimizer $\gamma^*(x, p)$ of the Hamiltonian $H$ is unique. Moreover, $H$ is strictly concave in $p$ and twice continuously differentiable with Lipschitz second derivatives. Finally, the costs $F$ and $g$ are continuously differentiable with Lipschitz derivatives, and both are **Lasry–Lions monotone** in the sense that for both $\phi = F, g$,*

$$\sum_{x \in [d]} (\phi(x, \eta) - \phi(x, \hat{\eta}))(\eta_x - \hat{\eta}_x) \geq 0, \qquad \eta, \hat{\eta} \in \mathcal{P}([d]). \quad (2.5)$$

We note that the first part of this assumption holds when $f$ is strictly convex in $a$. Additionally, Lasry–Lions monotonicity can be interpreted as the player's dislike for congestion (e.g., $\eta_x$ close to one). Under the assumptions outlined above, the forward-backward system in (2.3) attains a unique solution $(u^\eta, \mu^\eta)$, the MFG equilibrium. The argument proving existence follows from a fixed-point argument via Schauder's fixed-point theorem, while uniqueness results from Assumption 2.1. For more details, see (Carmona & Delarue, 2018a, Section 7.2.2), for instance.

## 2.2 Flow Maps and the Master Equation

We now turn to the question of solving the MFG for any initial distribution $\eta$. Although solving the MFG system (2.3) via Picard iteration for a given $\eta \in \mathcal{P}([d])$ is generally tractable, we aim to solve the system for all such $\eta$, and hence cannot rely solely on the MFG system.

**Master field.** We begin by considering the value function $u^\eta$, which solves the HJB equation in the MFG system (2.3) with initial distribution $\eta$. The value function depends implicitly on the mean field, and we make this dependence *explicit* by introducing the **master field** $U$, defined such that $U(t, x, \mu^\eta(t)) = u^\eta(t)$ for all $(t, x, \eta) \in [0, T] \times [d] \times \mathcal{P}([d])$. This object plays a central role in theory of MFGs and establishing a rigorous connection to finite-player games; see Bayraktar & Cohen (2018); Cecchin & Pelino (2019) and Appendix C for more details. The master field $U$ is also very relevant for applications: if the master field is known, then it can be evaluated along any flow of measures $\mu(t)$. Additionally, $U(t, x, \mu)$ is the optimal cost that a representative player can obtain if starting in state $x$ at time $t$, with the of the population starting in distribution $\mu$ and playing according to the equilibrium control.

Methods, such as Cohen et al. (2024), that learn $U$ by exploiting its connection with a nonlinear PDE called the master equation suffer from two limitations: **(1)** the computation of the loss function is complex and costly, and **(2)** they cannot handle situations where the terminal cost varies, as the loss function is defined in terms of a fixed terminal cost. For this reason, we develop a new approach, relying on the concept of flow map.

**Flow map.** Instead of focusing on the aforementioned master field, we will consider a function which maps the initial distribution and the terminal cost to the value function. In other words, we would like to learn the operator (since $g$ is a function)

$$\Phi : (t, \eta, g) \mapsto u^{\eta, g}(t), \quad (2.6)$$

where $u^{\eta,g}$ is the value function for the MFG system (2.3) with initial distribution $\eta \in \mathcal{P}([d])$ and terminal cost $g$. We recall that the control can be recovered from the value function using the relation $\widehat{\alpha}_y(t,x) = \gamma_x^*(y, \Delta_y u^{\eta,g}(t,\cdot))$. In turn, obtaining $\Phi$ concretely gives access to the MFG equilibrium for *any initial condition $\eta$ and terminal cost $g$.* In principle, the operator could be extended to include running costs and dynamics. We comment that, in line with the operator learning approach for HJB equations proposed in Lanthaler & Stuart (2025), the MFG system in Equation (2.3) can be viewed as the characteristics of the master field. In the same sense, our method is an operator learning method because we learn the characteristics of the master field to obtain its solution operator.

**Terminal cost parameterization.** When endowed with an appropriate norm, the set of all Lipschitz functions on the probability simplex is an infinite-dimensional Banach space. However, to obtain precise approximation and generalization guarantees, we restrict our attention to a *parameterized* class of terminal costs in this paper. Given a parameter $\kappa \in \mathbb{R}^k$, we denote by $g_\kappa$ the corresponding terminal cost function. Then, the flow map we focus on in the sequel is defined as follows.

**Definition 2.2.** *Given a parametrized family of terminal conditions, the **flow map** $\Phi : [0,T] \times \mathcal{P}([d]) \times \mathcal{K} \to \mathbb{R}^d$ is defined by $\Phi(t,\eta,\kappa) := u^{\eta,\kappa}(t)$, where $u^{\eta,\kappa}$ is the value function for the MFG system (2.3) with initial distribution $\eta \in \mathcal{P}([d])$ and terminal cost $g_\kappa$.*

We make two key remarks. First, the initial distribution $\eta$ and the parameter $\kappa$ may be high-dimensional, which justifies using neural networks to approximate $\Phi$. Second, contrary to the aforementioned master field $U$, the flow map $\Phi$ does not satisfy a PDE and hence it will require a novel training algorithm, based on the MFG equilibrium characterization.

We conclude with a regularity condition that allows the rigorous study of the approximation of $\Phi$ by neural networks in the next section. This assumption holds in the test cases consider below in our numerical experiments (see Section 5).

**Assumption 2.2.** *There exists a compact set of parameters $\mathcal{K} \subseteq \mathbb{R}^k$ such that for all $\kappa \in \mathcal{K}$, the $g_\kappa : [d] \times \mathcal{P}([d]) \to \mathbb{R}$ satisfies Assumption 2.1. Moreover, for any $\kappa, \kappa' \in \mathcal{K}$, there exists a constant $C > 0$ such that $|g_\kappa(x,\mu) - g_{\kappa'}(x,\mu)| \leq C|\kappa - \kappa'|$, uniformly in $(x,\mu) \in [d] \times \mathcal{P}([d])$.*

## 3 Algorithm to Learn Flow Maps for MFGs

In this section, we outline our algorithmic approach to learning MFG equilibria, motivated by the HJ-Net algorithm of Lanthaler & Stuart (2025). Recall that we aim to learn an approximation of the flow map $\Phi : [0,T] \times \mathcal{P}([d]) \times \mathcal{K} \to \mathbb{R}$ that maps a time, an initial condition, and a parameter $\kappa \in \mathcal{K}$ (corresponding to a terminal condition $g_\kappa$) to the value function $u^{\eta,\kappa}(t)$. As in (Lanthaler & Stuart, 2025, Section 4), learning the flow map requires sample trajectories. We approximate $\Phi$ by a neural network which is trained using samples consisting of $(t,\eta,\kappa)$ and the associated $u^{\eta,\kappa}(t)$.

**Sampling method.** We generate i.i.d. samples $(\eta,\kappa) \sim \rho$, where $\rho$ is a joint distribution on $\mathcal{P}([d]) \times \mathcal{K}$. Then, we compute $u^{\eta,\kappa}$. Since this value function is coupled with the flow of measures $\mu^{\eta,\kappa}$ that solves the MFG system (2.3), we solve this system by Picard iteration: given an initial guess, we alternatively solve the forward KFP equation and the backward HJB equation to update $\mu$ and $u$ respectively. We thus obtain an (approximate) solution of (2.3). In our implementation, we use a temporal finite-difference scheme with a mesh of $M$ steps, yielding an approximate solution $(\tilde{u}_i^{\eta,g}, \tilde{\mu}_i^{\eta,g})_{i=0,\dots,M}$. See Appendix D for additional details. We denote the Picard iteration map for an MFG with terminal condition $g$ by $\Gamma_g : \mathcal{P}([d]) \to (\mathbb{R}^d)^{M+1}$. Intuitively, $\Gamma_g : \eta \mapsto u^{\eta,g}$. In practice, $\Gamma_g(\eta)$ is the vector of values $\tilde{u}_j^{\eta,g} \approx u^{\eta,g}(jT/M, \cdot) \in \mathbb{R}^d$, $j = 0, \dots, M$.

**Architecture.** We approximate $\Phi$ by a neural network. Since our goal in the next section is to obtain theoretical guarantees, we focus here on a relatively simple architecture, but more complex architectures are explored in our numerical experiments. We limit ourselves to fully-connected ReLU neural networks $\phi : \mathbb{R}^{k_1} \to \mathbb{R}^{k_2}$ of depth $L$. Following the convention in Jiao et al. (2023), from which we derive our generalization guarantee, such networks are recursively defined by $\phi_0(x) = x$, $\phi_{j+1}(x) = \sigma(A_j \phi_j(x) + b_j)$ for $j = 1, \dots, L-1$, and $\phi(x) := A_L \phi_L(x)$. Above, the weights satisfy $A_j \in \mathbb{R}^{N_{j+1} \times N_j}$ for $j = 0, \dots, L$ and $b_j \in \mathbb{R}^{N_{j+1}}$ for $j = 0, \dots, L-1$, where $N_0 = k_1$ and $N_{L+1} = k_2$. By the width of a neural network, we refer to $W := \max\{N_1, \dots, N_L\}$, the maximum number of neurons in a hidden layer. For brevity, we denote such a network by $\phi(x; A, b)$, where $A = (A_0, \dots, A_{L-1})$ and $b = (b_0, \dots, b_{L-1})$.

**Training method.** We learn the flow map by training such a neural network on the samples generated by Picard iteration. To alleviate the notation, we denote $x = (jT/M, \eta, \kappa)$ and $y = \tilde{u}_j^{\eta,\kappa}$, where $j \in [M]$ and we recall that $\tilde{u}^{\eta,\kappa}$ is the discrete time approximation of the value function $u^{\eta,\kappa}$. Given samples $\{x_i, y_i\}_{i=1}^n$ from the procedure outlined above, we minimize the empirical loss

$$\mathcal{L}_n(A, b; \{x_i, y_i\}_{i=1}^n) := \frac{1}{n} \sum_{i=1}^n \ell(\phi(x_i; A, b), y_i), \tag{3.1}$$

where $\ell : \mathbb{R}^d \times \mathbb{R}^d \to \mathbb{R}$ is a convex loss and the minimum is taken over $A = \{A_j\}_{j=0}^L$ and $b = \{b_j\}_{j=0}^{L-1}$ simultaneously. In practice, this is accomplished using batch stochastic gradient descent (SGD) with a standard optimizer such as AdamW (Loshchilov & Hutter, 2019). If $(A^*, b^*) := \arg\min_{A,b} \mathcal{L}_n(A, b; \{x_i, y_i\}_{i=1}^n)$ (noting that these parameters depend on the sampled trajectories), we define our approximate flow map $\Psi_n(t, \eta, \kappa) := \phi(t, \eta, \kappa; A^*, b^*)$. This procedure is summarized in Algo. 1 below (written using SGD as the optimizer for simplicity).

---

**Algorithm 1** Sampling and Learning Flow Map for a Family of MFGs

---

**Input:** Number of time steps $M \in \mathbb{N}$, parameter set $\mathcal{K} \subset \mathbb{R}^k$, number of samples $n \in \mathbb{N}$, Picard solver $\Gamma$, number of training steps $m_{\text{train}}$, mini-batch size $n_{\text{mini}} < n$, learning rate $\{\gamma_j\}_{j \in \mathbb{N}}$
  1: Sample $\{(\eta_i, \kappa_i)\}_{i=1}^n$ uniformly and independently in $\mathcal{P}([d]) \times \mathcal{K}$
  2: **for** $i = 1, \dots n$ **do** ▷ Sample generation via Picard iteration
  3:     $\tilde{u} \leftarrow \Gamma_{g_{\kappa_i}}(\eta_i)$
  4:     Draw $j \sim \text{Unif}([M])$
  5:     $x_i \leftarrow (jT/M, \eta_i, \kappa_i)$
  6:     $y_i \leftarrow \tilde{u}_j$
  7: Initialize neural network parameters $(A^{(0)}, b^{(0)})$
  8: **for** $j = 1, \dots, m_{\text{train}}$ **do** ▷ Train neural network approximator
  9:     Sample mini-batch $\{(x_i, y_i)\}_{i=1}^{n_{\text{mini}}}$ from $\{(x_i, y_i)\}_{i=1}^n$
 10:     $(A^{(j)}, b^{(j)}) \leftarrow (A^{(j)}, b^{(j)}) - \gamma_j \nabla_{A,b} \mathcal{L}_{n_{\text{mini}}}(A, b; \{x_i, y_i\}_{i=1}^{n_{\text{mini}}})$ ▷ Gradient step
 11: **return** $\widehat{\Psi}_n(t, \eta, \kappa) = \phi(t, \eta, \kappa; A^{(m_{\text{train}})}, b^{(m_{\text{train}})})$

---

# 4 THEORETICAL GUARANTEES

We provide the following **two theoretical guarantees** for our proposed approach:

(1) **Approximation error (Corollary 4.3):** There exists a ReLU neural network approximating the true flow map $\Phi$ with error $\mathcal{O}(K^{-1/(d+k+2)})$, width $W = \mathcal{O}(K^{(2(d+k)+3)/(2(d+k)+4)})$, and depth $L = \mathcal{O}(\log(d + k + 1))$, all quantified in terms of a bound $K \geq 1$ on the weights of the network, the number of states $d$ of the underlying family of MFGs, and the dimension $k$ of the set that parametrizes the family of MFGs.

(2) **Generalization error (Corollary 4.5):** Learning the flow map via empirical risk minimization with $n$ samples yields a neural network approximation with expected excess risk $\mathcal{O}(n^{-1/(d+k+4)} \log(n))$, up to any error from the optimization process.

These results rely on a preliminary regularity result about the regularity of the flow map $\Phi$ that we establish in Appendix E:

**Theorem 4.1.** *Under Assumptions 2.1 and 2.2, the flow map $\Phi : [0, T] \times \mathcal{P}([d]) \times \mathcal{K} \to \mathbb{R}^d$, given by $\Phi(t, \eta, \kappa) = u^{t_0, \eta, \kappa}(t, \cdot)$, is jointly Lipschitz in its inputs: there exists $C > 0$ such that*

$$|\Phi(t, \eta_1, \kappa_1) - \Phi(s, \eta_2, \kappa_2)| \leq C(|t - s| + |\eta_1 - \eta_2| + |\kappa_1 - \kappa_2|)$$

*for all $(t, \eta_1, \kappa_1), (s, \eta_2, \kappa_2) \in [0, T] \times \mathcal{P}([d]) \times \mathcal{K}$.*

The approach of Lanthaler & Stuart (2025), which relies on (Yarotsky, 2017, Theorem 1), cannot be used in our case (see Rem. E.1). Instead, we develop an alternative analysis building upon Jiao et al. (2023). Given a ReLU neural network with weight matrices $\{A_j\}_{j=0}^L$ and biases $\{b_j\}_{j=0}^{L-1}$, let

$$p(\{A_j\}, \{b_j\}) := \|A_L\| \prod_{j=0}^{L-1} \max\{\|(A_j, b_j)\|, 1\}.$$

Then, the set of neural networks with width $W$, depth $L$, and norm bound satisfying $p(\{A_j\}, \{b_j\}) \leq K$ is denoted by $\mathcal{NN}(W, L, K)$. In this class of neural networks, (Jiao et al., 2023, Theorem 3.2) provides the following approximation result. Below, the space of functions $\mathcal{C}^{0,1}([0,1]^d)$ refers to the space of Lipschitz continuous functions on $[0,1]^d$.

**Proposition 4.2.** *There exists constants $c, C > 0$ such that for any $K \geq 1$, $W \geq cK^{(2d+1)/(2d+2)}$, and $L \geq 2\lceil\log(d)\rceil + 2$, the worst-case approximation error of the class $\mathcal{NN}(W, L, K)$ for $\Phi \in \mathcal{C}^{0,1}([0,1]^d)$ satisfies:* $\sup_{\Phi \in \mathcal{C}^{0,1}([0,1]^d)} \inf_{\Psi \in \mathcal{NN}(W,L,K)} \|\Phi - \Psi\|_{\mathcal{C}([0,1]^d)} \leq CK^{-1/(d+1)}$.

More concisely, over the class $\mathcal{C}^{0,1}([0,1]^d)$ of Lipschitz functions, the worst-case approximation error with a sufficiently wide and deep ReLU neural network can be quantified precisely in terms of a bound on the weights of the approximating networks. Using Theorem 4.1, we obtain the following as a corollary, with proof in Appendix E:

**Corollary 4.3.** *Assume that Assumptions 2.1 and 2.2 hold. Then, for any $K \geq 1$ and $\varepsilon > 0$, there exists a neural network $\Psi \in \mathcal{NN}(W, L, K)$ with weight bound $K$, width $W \geq c(\mathrm{diam}(\mathcal{K}), T)dK^{(2(d+k)+3)/(2(d+k)+4)}$, and depth $L \geq 2\lceil\log(d + k + 1)\rceil + 2$ such that*

$$\|\Phi - \Psi\|_{\mathcal{C}([0,T] \times \mathcal{P}([d]) \times \mathcal{K})} \leq C(\mathrm{diam}(\mathcal{K}), T)K^{-1/(d+k+2)} + \varepsilon,$$

*where $\Phi : [0,T] \times \mathcal{P}([d]) \times \mathcal{K} \to \mathbb{R}^d$ is the flow map from Definition 2.2.*

We note that the results in Jiao et al. (2023) are presented in the setting of scalar-valued functions. In Appendix E, we show how we extend to the vector-valued setting that we require for our particular flow map. Such a result is particularly useful because of the generalization guarantees that arise from Rademacher complexity estimates for families of neural networks with bounded weights. For instance, (Jiao et al., 2023, Theorem 4.1) provides such a guarantee, in the context of regression, while (Jiao et al., 2023, Corollary 4.2) provides an analogous guarantee for noiseless regression problems with regularization. Suppose that we have $n$ samples $\{(x_i, y_i)\}_{i=1}^n$ such that $x_i \overset{\text{i.i.d.}}{\sim} \rho$, a distribution supported on $[0,1]^d$, and $y_i = \Phi(x_i)$ with $i = 1, \ldots, n$, where $\Phi : [0,1]^d \to \mathbb{R}$ belongs to $\mathcal{C}^{0,1}([0,1]^d)$ (i.e., it is Lipschitz continuous). Then, given fixed widths, depths, and weight bounds $W, L, K > 0$, the empirical risk is given exactly as in Equation (3.1), and and the empirical risk minimizer is

$$\Psi_n := \underset{\Psi \in \mathcal{NN}(W,L,K)}{\arg\min} \mathcal{L}_n(\Psi; \{(x_i, y_i)\}_{i=1}^n). \tag{4.1}$$

We take as our convex loss $\ell(x, y) = \|x - y\|_2^2$ for simplicity, as in (Jiao et al., 2023). Note that, for each $\Psi \in \mathcal{NN}(W, L, K)$, this quantity provides an unbiased estimate of the population risk $\mathcal{L}(\Psi) := \mathbb{E}_{x_i \sim \rho}[\ell(x_i, \Psi(x_i))]$. Now, suppose that we have computed the empirical risk minimizer in (4.1), up an optimization error $\varepsilon_{\mathrm{opt}} > 0$, via stochastic gradient descent, yielding a neural network $\widehat{\Psi}_n$ that satisfies

$$\mathcal{L}_n(\widehat{\Psi}_n) \leq \inf_{\Psi \in \mathcal{NN}(W,L,K)} \mathcal{L}_n(\Psi) + \varepsilon_{\mathrm{opt}}. \tag{4.2}$$

Then, we aim to quantify the excess risk, defined as $\|\widehat{\Psi}_n - \Phi\|_{L^2(\rho)}^2 := \mathcal{L}(\widehat{\Psi}_n) - \mathcal{L}(\Phi)$. A standard computation then shows that the expected excess risk, with expectation taken over the samples $\{x_i\}_{i=1}^n$, is given by

$$\mathbb{E}[\|\widehat{\Psi}_n - \Phi\|_{L^2(\rho)}^2] \leq \inf_{\Psi \in \mathcal{NN}(W,L,K)} \|\Psi - \Phi\|_{L^2(\rho)}^2 + \mathbb{E}[\mathcal{L}(\widehat{\Psi}_n) - \mathcal{L}_n(\widehat{\Psi}_n)] + \varepsilon_{\mathrm{opt}}.$$

To quantify the expected excess risk, it suffices to quantify the approximation error and the generalization error, the first and second terms in the above bound respectively. (Jiao et al., 2023, Theorem 4.1) combines Proposition 4.2 and a symmetrization argument to show the following:

**Proposition 4.4.** *If $\Phi \in \mathcal{C}^{0,1}([0,1]^d)$, then there exists $\tilde{C} > 0$ such that for $K = \mathcal{O}(n^{(d+1)/(2d+6)})$, $W \geq \tilde{C}K^{(2d+1)/(2d+2)}$, $L \geq 2\lceil\log(d)\rceil + 3$, any neural network $\widehat{\Psi}_n \in \mathcal{NN}(W, L, K)$ satisfying (4.2) also satisfies:* $\mathbb{E}[\|\widehat{\Psi}_n - \Phi\|_{L^2(\rho)}^2] - \varepsilon_{\mathrm{opt}} \leq \tilde{C}n^{-1/(d+3)}\log(n)$.

In general, it is difficult to quantify the optimization error $\varepsilon_{\mathrm{opt}}$. However, with sufficient hyperparameter tuning to stabilize training, we can safely assume that $\varepsilon_{\mathrm{opt}}$ is small. To conclude, Proposition 4.4 applies nearly *verbatim* in our setting, up to a rescaling argument found in Appendix E:

**Corollary 4.5.** *If $K = \mathcal{O}(n^{(d+k+2)/(2(d+k)+8)})$, then under the assumptions of Corollary 4.3, minimizing the empirical loss in Equation (3.1) over $n$ samples (generated via Algo. 1) yields a neural network $\widehat{\Psi}_n$ that satisfies, up to an optimization error $\varepsilon_{\text{opt}} > 0$, $\mathbb{E}[\|\widehat{\Psi}_n - \Phi\|^2_{L^2(\rho)}] - \varepsilon_{\text{opt}} \leq \tilde{C}(\text{diam}(\mathcal{K}), T)n^{-1/(d+k+4)} \log(n)$. Above, $\rho$ is the uniform distribution over $[0, T] \times \mathcal{P}([d]) \times \mathcal{K}$.*

## 5 NUMERICAL EXPERIMENTS

In this section, we provide numerical evidence for the accuracy and generalization of our method on two standard examples of finite-state MFGs. First, we demonstrate our scheme's accuracy on a simple cybersecurity model in dimension $d = 4$. Then, we consider high-dimensional quadratic MFGs, illustrating that our approach maintains its accuracy as the dimension of the underlying family of MFGs increases. Full experimental details are in Appendix A.

**Example 1: Low-Dimensional Cybersecurity Model.** We begin with a cybersecurity model introduced by Kolokoltsov & Bensoussan (2016) and studied in (Cohen et al., 2024, Section 7.4). Players can either protect or defend their computers against infection by malware. Before passing to the mean-field limit, each player can either be infected by a hacker or by interacting with another infected player. The player is either defended or undefended ($D$ or $U$) and susceptible or infected ($S$ or $I$), leading to a state space with $d = 4$ states: $\{DS, DI, US, UI\}$. The player determines whether to defend or not with a switching parameter $\rho > 0$, and the player pays cost $k_D > 0$ for defending and $k_I > 0$ if they are infected. The running cost is $f(x, a) = k_D \mathbb{1}_{\{DS, DI\}}(x) + k_I \mathbb{1}_{\{DI, UI\}}(x)$, and $F(x, \eta) \equiv 0$. The player's control is simply $a \in \{0, 1\}$, and this yields a transition matrix exactly as in (Cohen et al., 2024, Section 7.4). Importantly, we modify the original example by including a terminal cost, penalizing infected players at the terminal time $T$ according to a parameter $\kappa \geq 0$: $g_\kappa(x, \eta) = \kappa \mathbb{1}_{\{DI, UI\}}(x)$. We use Algo. 1 with $n = 2000$ samples, $m_{\text{train}} = 2000$ epochs with batches of size $m_{\text{mini}} = 64$. After training the neural network, we evaluate it on several pairs $(\eta, \kappa)$ to obtain $\widehat{u}$ and compare with the solution obtained by solving the ODE system with this pair of initial and terminal conditions. Fig. 2 shows that our method performs well on random samples with $\kappa \in [0, 10]$ and arbitrary $\eta \in \mathcal{P}([4])$. Additionally, having learned the value function $u$, we can easily recover the flow of measures $\mu$ by simply solving the KFP equation with the learned value function. The results of such experiments are displayed in Figure 3, showing that our method also allows for the accurate recovery of the flow of measures at the MFG equilibrium. Appendix G contains more experiments with this model, including an illustration of the case that $\kappa = 0$ (i.e., the setting considered in Cohen et al. (2024)).

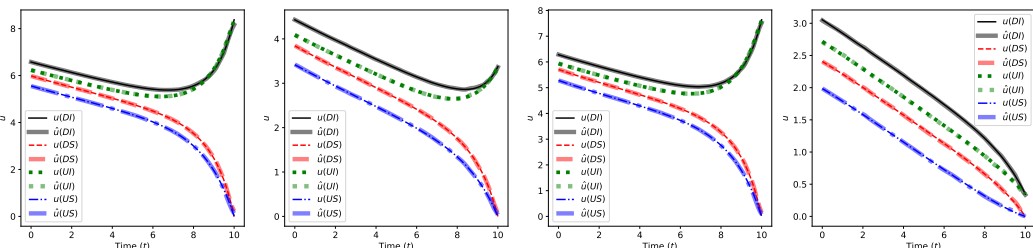

Figure 2: Learned value function $\widehat{u}$ and true value function $u$ for four random initial distributions $\eta$ and final cost parameter $\kappa \in [0, 10]$, both drawn uniformly at random from $\mathcal{P}([4])$ and the interval $[0, 10]$, respectively. Each curve corresponds to one state in $\{DS, DI, US, UI\}$.

**Example 2: High-Dimensional Quadratic Models.** We also consider the quadratic cost model, as in (Cohen et al., 2024, Section 7.1), also analyzed in (Cecchin & Pelino, 2019, Example 1) and (Bayraktar & Cohen, 2018, Example 3.1) via the master equation. This setting allows us to test our method on high-dimensional MFGs, and the assumptions that we impose on the parametrized family of terminal costs remain easily verifiable. We take a quadratic running cost and a linear mean-field cost, given by $f(x, a) := b \sum_{y \neq x}(a_y - 2)^2$, $F(x, \eta) := \eta_x$, with action space $\mathbb{A} := [1, 3]$ and $b = 4$. As shown in (Cohen et al., 2024, Section 7.1), letting $T = 1$ will ensure that the resulting Hamiltonian satisfies our assumptions. Therein, the authors take $g(x, \eta) \equiv 0$, but we convert their quadratic model into a parametrized family of MFGs by taking instead $\kappa \in [0, 1]^d =: \mathcal{K}$ and

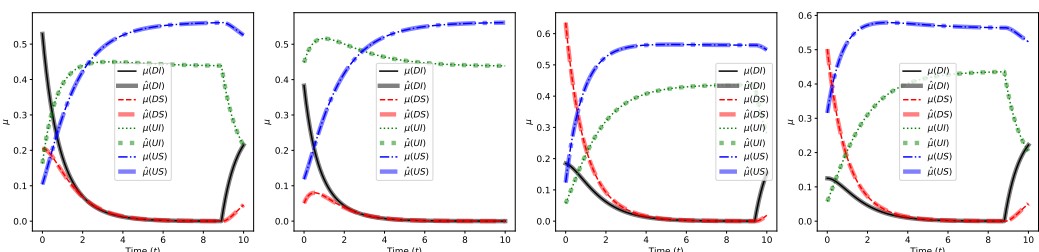

Figure 3: Learned flow of measures $\widehat{\mu}$ and true flow of measures $\mu$ for four random initial distributions $\eta$ and final cost parameter $\kappa \in [0, 10]$, both drawn uniformly at random from $\mathcal{P}([4])$ and the interval $[0, 10]$, respectively.

considering terminal costs of the form $g_\kappa(x, \eta) = \kappa_x + \eta_x$. The inclusion of $\kappa$ in the terminal cost has the effect of pushing the player away from states $x \in [d]$ such that $\kappa_x$ is large and towards states with small $\kappa_x$, with the mean-field term $\eta_x$ discouraging crowding. As our numerical results demonstrate, the value function depends heavily on the parameter $\kappa$, making this a challenging task, especially as $d$ increases.

In Fig. 4, we demonstrate the success of our method in learning the flow map for this family of MFGs in dimensions $d = 3$, $d = 4$, $d = 5$, and $d = 10$ respectively. Additionally, Fig. 5 displays the errors, measured as the difference between the true and approximate value function at each time, in the preceding figure. Beyond dimension $d = 10$, learning becomes increasingly unstable, as the number of samples required to learn to high precision becomes intractable to generate in a reasonable amount of time using our own computational resources, in line with how the sample complexity estimate from Corollary 4.5 scales with dimension $d$. However, in Appendix G, we show that by passing to a *time* discretization, our method still generalizes well to dimension $d = 20$. Therein, we also provide evidence that using a neural network architecture with skip connections and layer normalization (e.g., ResNet) can improve training stability in dimension $d = 10$. While the optimization procedure is still relatively stable in dimensions $d = 3, 4, 5, 10$ with feedforward ReLU networks, we anticipate that adding skip connections smooths out the loss landscape and helps prevent our method from getting trapped in spurious local minima. This observation is supported by the theoretical and empirical evidence from Balduzzi et al. (2017), although we remark that, even with feedforward ReLU networks, our method performs well in dimension $d = 10$, as in Figure 5d.

Finally, Fig. 6, we illustrate both the training and test loss over the course of Algo. 1 for the quadratic model in dimension $d = 3$. Averaging over five trials, we provide empirical evidence for both Corollary 4.3 and Corollary 4.5, showing that increasing width results in models that **(1)** learn the flow map to greater accuracy (Fig. 6a) and **(2)** generalize better to unseen samples (Fig. 6b).

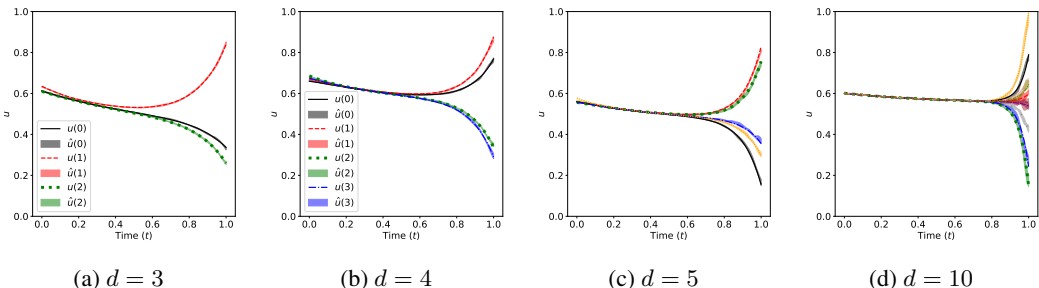

(a) $d = 3$      (b) $d = 4$      (c) $d = 5$      (d) $d = 10$

Figure 4: Comparison of true value functions $u$ and learned value functions $\widehat{u}$ for randomly sampled pairs $(\eta, \kappa)$ in dimensions $d = 3, 4, 5, 10$ respectively. Averages are taken across 5 trials, and shaded regions on approximate curves indicate error bars of one standard deviation, computed across trials.

In Table 1 and Table 2 in Appendix A, we describe the optimizers and loss functions that we chose for each of our experiments, including the additional experiments in Appendix G. In order to verify our theoretical results, which are restricted to the setting of feedforward ReLU networks, we primarily

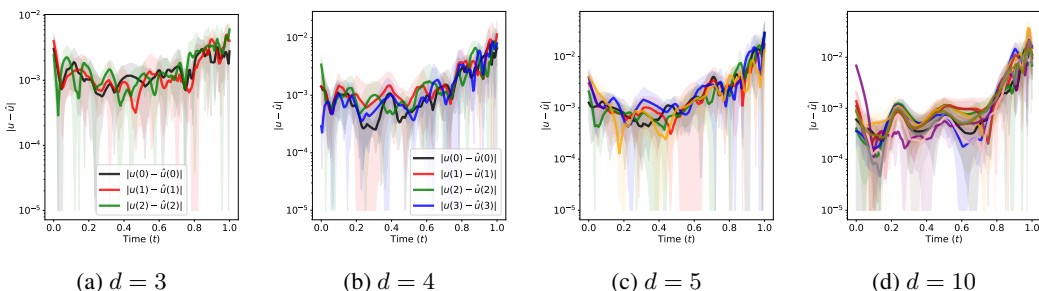

(a) $d = 3$      (b) $d = 4$      (c) $d = 5$      (d) $d = 10$

Figure 5: Absolute errors of learned value functions $|u - \widehat{u}|$ for randomly sampled pairs $(\eta, \kappa)$ in dimensions $d = 3, 4, 5, 10$ respectively. Averages are taken across 5 trials, and shaded region s on approximate curves indicate error bars of one standard deviation, computed across trials.

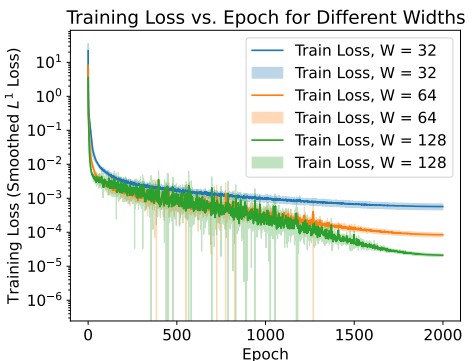 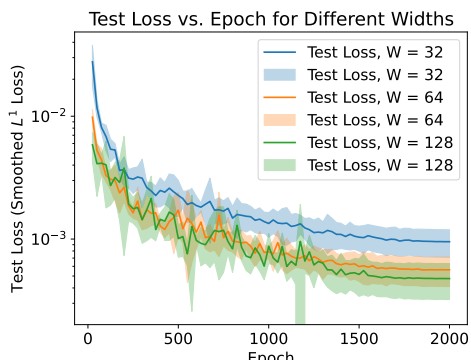

(a) Training loss vs. epoch for $d = 3$ dimensional quadratic model and $W = 32, 64, 128$.

(b) Test loss vs. epoch for $d = 3$ dimensional quadratic model and $W = 32, 64, 128$.

Figure 6: Comparison of training loss and test loss, evaluated on held-out data every 25 epochs, for ReLU neural networks with width $W = 32, 64, 128$, depth $L = 4$, and $n = 4000$ samples. Shaded regions represent one standard deviation above/below the mean of five trials. As $W$ increases, the optimization procedure becomes more unstable but both the training and test losses decrease.

present experiments for this architecture. When selecting an optimizer, we observe that Adam and AdamW perform comparably, although for the high-dimensional quadratic model, AdamW's performance is slightly better. We also find that smooth $L^1$-loss, as opposed to $L^2$-loss, provides a more stable optimization trajectory in the high-dimensional setting. Overall, the optimization of feedforward ReLU networks in the high-dimensional setting is more challenging, but standard optimization tools still yield accurate results *without* relying on more complex neural network architectures.

## 6 CONCLUSION

We present an operator learning method for solving parametrized families of finite-state MFGs. To our knowledge, our approach provides the most general learning-based framework for solving finite-state MFGs. Our theoretical guarantees rigorously quantify the approximation error, parametric complexity, and generalization performance, and our numerical experiments illustrate the empirical accuracy of our method for a variety of common finite-state MFGs. Our method extends naturally to MFGs with parametrized running costs, with only slight modifications to our regularity proofs required and no modification to Algo. 1. We believe that our sampling algorithm, although intuitive, could be improved to gain greater coverage of the flow map's domain, allow for more stable optimization, and enable better generalization. Techniques such as oversampling in regions with poor coverage or adversarial training may prove beneficial. Future work will also include extending our results to continuous state-space MFGs and infinite-dimensional spaces of cost functions, for which powerful operator learning architectures (e.g., DeepONets or FNO) will likely be instrumental.

**Reproducibility Statement.** We include a detailed description of our numerical experiments, including computational resources used, training methodology, and hyperparameters for all experiments in Appendix A. Additionally, we have submitted all code used for experiments presented in Section 5 and Appendix G as supplementary material. For our theoretical results, all assumptions are provided in Section 2.1 and expanded upon in Appendix B, while our technical proofs can all found be found in Appendix E.

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

## A    EXPERIMENTAL DETAILS

As noted in our reproducibility statement, we provide all experimental details in order to recreate our results from Section 5 and in Appendix G. Smaller experiments in the cybersecurity example were carried out a 2020 MacBook Pro with an Apple M1 chip and 8GB RAM. For the purposes of timing (see Appendix G), higher-dimensional experiments in the quadratic example were instead run on a single NVIDIA A100 TensorCore GPU with 40GB of VRAM via Google Colaboratory. All experiments were implemented in the PyTorch machine learning library in Python (Paszke et al., 2019). Code for our numerical experiments can be found in our submitted supplementary material.

To best align with our theoretical results in Section 4, we utilize fully-connected ReLU neural networks for all experiments unless otherwise specified. In all cases, we used mini-batches with $n_{\mathrm{mini}} = 64$ samples for each gradient step during the training loop. All hyperparameters were selected via a grid search, with tuned parameters being: initial learning rate, number of hidden layers (depth), width of each hidden layer, number of training epochs, and number of training samples. In each case, we validated our models by testing on 20% of the training data, held-out from the training set for validation. In many cases, we found it beneficial to utilize early stopping to prevent overfitting; for higher-dimensional examples, training for *fewer* epochs appears to provide better results. Finally, in all cases, we found that a cosine annealing learning rate scheduler performed best for optimization; we used the default parameters for the CosineAnnealingLR scheduler, as implemented in PyTorch's torch.optim package. We also found that optimization was more slightly stable for higher-dimensional cases when using: (1) the AdamW optimizer with the default weight decay parameter, $\lambda = 0.01$, and (2) smooth $L^1$-loss in place of $L^2$-loss. Nonetheless, we are able to obtain similar results with $L^2$-loss, in line with our theoretical framework in Section 4.

In Tables 1 and 2 below, we describe the specific architectures and hyperparameters that we chose for each experiment in Section 5, including depth, width, number of training epochs, optimizer parameters, and number of training samples. Finally, in Appendix G, we provide additional numerical

experiments to showcase the accuracy of our method for higher-dimensional quadratic models. Departing from feedforward ReLU networks, we are able to obtain even better performance in $d = 10$ using a ResNet architecture with depth $L = 4$, two layers of width 128, two hidden layers with width 64, skip connections between all layers, layer normalization, and dropout with probability $p = 0.05$. See the results of this experiment in Fig. 7 below.

Our submitted code, provided in the supplementary material, is organized as follows:

- `utils` contains a generic class for MFG operators, as well as two scripts defining the cybersecurity and quadratic models from Section 5.

- `generation` contains two scripts for sampling trajectories from the cybersecurity and quadratic models, respectively.

- `tests` contains

    - `train_cs_operator_time.py` and `train_quad_operator_time.py`, which implement Algorithm 1 for the two models in Section 5, and

    - `train_cs_operator_fixed.py` and `train_quad_operator_fixed.py`, which implement a similar algorithm for learning trajectories along a *fixed* time discretization.

- Finally, the directories `data`, `models`, and `plots` contain example outputs that can be reproduced (at least, up to randomness of sampling) by running the above scripts.

For example, running `generate_quad_data.py` will generate samples for the quadratic model in dimension $d = 3$, stored in `data` then running `train_quad_operator_time.py` will train a model on the generated samples, outputting a model stored in `models` and a corresponding plot on four random samples, stored in `plots`.

Table 1: Optimization details for experiments Section 5 and Appendix G.

| Experiment | Optimizer | Loss Function |
|---|---|---|
| Cybersecurity Model | Adam | $L^2$ |
| Cybersecurity Model (Fixed Discretization) | Adam | $L^2$ |
| Quadratic Model | AdamW | Smooth $L^1$ |
| Quadratic Model (Fixed Discretization) | Adam | $L^2$ |

Table 2: Selected hyperparameters for experiments in Section 5 and Appendix G.

| Experiment | # Training Samples ($n$) | # Epochs ($m_{\text{train}}$) | Width ($W$) | Depth ($L$) | Initial Learning Rate |
|---|---|---|---|---|---|
| Cybersecurity Model | 2000 | 2000 | 64 | 4 | $8 \times 10^{-4}$ |
| Cybersecurity Model (Fixed Discretization) | 2000 | 1000 | 64 | 4 | $8 \times 10^{-4}$ |
| Quadratic Model ($d = 3$) | 4000 | 2000 | 64 | 4 | $8 \times 10^{-4}$ |
| Quadratic Model ($d = 3$, Fixed Discretization) | 4000 | 1000 | 64 | 4 | $8 \times 10^{-4}$ |
| Quadratic Model ($d = 4$) | 4000 | 2000 | 64 | 4 | $8 \times 10^{-4}$ |
| Quadratic Model ($d = 4$, Fixed Discretization) | 4000 | 1000 | 64 | 4 | $8 \times 10^{-4}$ |
| Quadratic Model ($d = 5$) | 4000 | 2000 | 64 | 4 | $8 \times 10^{-4}$ |
| Quadratic Model ($d = 5$, Fixed Discretization) | 4000 | 1000 | 64 | 4 | $8 \times 10^{-4}$ |
| Quadratic Model ($d = 10$) | 10000 | 500 | 128 | 4 | $1 \times 10^{-4}$ |
| Quadratic Model ($d = 10$, Fixed Discretization) | 10000 | 1000 | 64 | 4 | $8 \times 10^{-4}$ |
| Quadratic Model ($d = 20$, Fixed Discretization) | 20000 | 1000 | 64 | 4 | $8 \times 10^{-4}$ |

## B  MARKOVIAN CONTROLS AND REPRESENTATIVE PLAYER'S PROCESS

In this appendix, we formally describe the Markovian controls that the representative player in a finite-state MFG chooses, presented at a high level in Section 2.1.

Denoting $[d] = \{1, \ldots, d\}$ to be the set of states that the player may switch between, a Markovian control refers to a measurable function

$$\alpha : \mathbb{R}_+ \times \{1, \ldots, d\} \to \mathbb{A}_{[d]}^d = \bigcup_{x \in [d]} \mathbb{A}_{-x}^d,$$

where
$$\mathbb{A}^d_{-x} := \left\{ a \in \mathbb{R}^d : \forall y \neq x, \quad a_y \in \mathbb{A}, \quad a_x = -\textstyle\sum_{y \neq x} a_y \right\}.$$

The value of $\alpha_y(t,x) := \alpha(t,x)_y$, where $x \neq y$, represents the player's rate of transition at time $t$ from the state $x$ to the state $y$. We require that $\alpha_x(t,x) = -\sum_{y \neq x} \alpha_y(t,x)$ for all $x \in [d]$, as is standard for the transition probabilities of a continuous-time Markov chain. More concisely, let $\mathcal{Q}[\mathbb{A}]$ be the set of $d \times d$ transition-rate matrices with rates in $\mathbb{A} := [\mathfrak{a}_l, \mathfrak{a}_u]$. Then, the player chooses Markovian controls $\alpha : [0,T] \to \mathcal{Q}[\mathbb{A}]$ which we refer to as the set of admissible controls. Under this interpretation, $(\alpha(t))_{x,y} = \alpha_y(t,x)$.

In Section 2.1, we noted that, given a Markovian control $\alpha$ and an initial distribution $\eta \in \mathcal{P}([d])$, the player's dynamics obey a continuous-time Markov chain with transition probabilities
$$\Pr(\mathcal{X}^\eta_{t+h} = y \mid \mathcal{X}^\eta_t = x) = \alpha_y(t,x)h + o(h), \quad h \to 0^+.$$

In fact, this Markov chain arises as the result of a Poisson jump process, which completely describes the dynamics of the representative player. Our method does not rely on the exact details of the jump process, however, and we thus refer the interested reader to (Cecchin & Fischer, 2018, Section 2.3) for additional details on the probabilistic structure of finite-state MFGs that we consider.

In Section 2.1, we provided a condensed version of the assumptions that ensure that the MFG system has a unique solution. Below, we expand on these assumptions, providing the full suite of assumptions that previous work such as Bayraktar & Cohen (2018); Cecchin & Pelino (2019); Cohen et al. (2024) all utilize.

Our first two assumptions ensure that the Hamiltonian in (2.4) has a unique minimizer and that the running and terminal costs $F$ and $g$ are monotone in an appropriate sense. Our third assumption is a technical assumption on the strong concavity of the Hamiltonian. Although this last assumption may not appear immediately relevant, it is useful later when we analyze the regularity of the flow map for Equation (2.3) in Appendix E.1 below.

**Assumption B.1.** *The Hamiltonian has a unique minimizer, which we refer to as the* optimal rate selector *and is denoted* $\gamma^*(x,p) := \arg\min_{a \in \mathbb{A}^d_{-x}} \{f(x,a) + a \cdot p\}$. *The optimal rate selector* $\gamma^*$ *is a measurable function that, given any* $(x,p) \in [d] \times \mathbb{R}^d$, *defines a well-defined (unique) rate vector* $a$ *such that for any* $y \neq x$, $a_y \in \mathbb{A}$ *and* $a_x = -\sum_{y \neq x} a_y$. *In particular, it is sufficient that* $f$ *is strictly convex with respect to* $a$.

**Assumption B.2.** *The functions* $F$ *and* $g$ *are continuously differentiable in* $\eta$ *with Lipschitz derivatives. Moreover,* $F$ *and* $g$ *are Lasry–Lions monotone in the sense that for both* $\phi = F, g$,
$$\sum_{x \in [d]} (\phi(x,\eta) - \phi(x,\hat\eta))(\eta_x - \hat\eta_x) \geq 0, \tag{B.1}$$
*for any* $\eta, \hat\eta \in \mathcal{P}([d])$.

**Assumption B.3.** *Assume that, for some* $W > 0$, *the derivatives* $D^2_{pp}H$ *and* $D_pH$ *of the Hamiltonian exist and are Lipschitz in* $p$ *on* $[-W, W]$. *Moreover,* $H$ *is strictly concave in* $p$: *there exists a positive constant* $C_{2,H}$ *such that:*
$$D^2_{pp}H(x,p) \leq -C_{2,H}. \tag{B.2}$$

When $H$ is differentiable, (Gomes et al., 2013, Proposition 1) implies that
$$\gamma^*(x,p) = D_pH(x,p), \tag{B.3}$$

a useful property when establishing regularity of the flow map. Moreover, if Assumption B.3 holds, then $\gamma^*$ is locally Lipschitz.

## C  THE MASTER EQUATION

The master equation is given by the following nonlinear PDE:
$$\begin{cases} \partial_t U(t,x,\eta) + \sum_{y,z \in [d]} \eta_y D^\eta_{yz} U(t,x,\eta)\gamma^*_z(y, \Delta_y U(t,\cdot,\eta)) + \bar{H}(x,\eta, \Delta_x U(t,\cdot,\eta)) = 0, \\ U(T,x,\eta) = g(x,\eta), \qquad (t,x,\eta) \in [0,T] \times [d] \times \mathcal{P}([d]), \end{cases}$$
$$\tag{C.1}$$

Above, $U : [0, T] \times [d] \times \mathcal{P}([d]) \to \mathbb{R}$, with $D_{yz}^{\eta}$ denoting a directional derivative in the direction of the vector $e_{yz} := e_y - e_z$ on the probability simplex, where $e_y, e_z \in \mathbb{R}^d$ are standard basis vectors indexed by $y, z \in [d]$. More precisely, for $\phi : \mathcal{P}([d]) \to \mathbb{R}$, we define

$$D_{yz}^{\eta}\phi(\eta) := \lim_{h \to 0^+} \frac{\phi(\eta + e_{yz}h) - \phi(\eta)}{h}. \tag{C.2}$$

Note that this convention respects the geometry of the simplex, in the sense that derivatives are only allowed in directions along the simplex: if $\eta \in \mathcal{P}([d])$, then $\eta + e_{yz}h \in \mathcal{P}([d])$ for all $h$ sufficiently small.

We have the following result concerning the master equation, both providing its regularity and establishing the consistency relation invoked in Corollary 4.1 above. This proposition combines results from (Cecchin & Pelino, 2019, Proposition 1, Proposition 5, Theorem 6) and (Cardaliaguet et al., 2019, Section 1.2.4).

**Proposition C.1.** *There exists a unique solution, denoted by $(u^{t_0, \eta}, \mu^{t_0, \eta})$, in $\mathcal{C}^1([t_0, T] \times [d], \mathbb{R}) \times \mathcal{C}^1([t_0, T] \times [d], \mathcal{P}([d]))$ to (2.3). Let $U$ be defined by:*

$$U(t_0, x, \eta) := u^{t_0, \eta}(t_0, x). \tag{C.3}$$

*Then, the master field $U$ is the unique classical solution to the master (C.1). Moreover, we have the consistency relation such that for all $t_0 \in [0, T]$,*

$$U(t, x, \mu^{t_0, \eta}(t)) = u^{t, \mu^{t_0, \eta}(t)}(t) = u^{t_0, \eta}(t), \qquad (t, x, \eta) \in [t_0, T] \times [d] \times \mathcal{P}([d]). \tag{C.4}$$

*Finally, $U(\cdot, x, \cdot) \in \mathcal{C}^{1,1}([0, T] \times \mathcal{P}([d]))$ for every $x \in [d]$.*

Note that the above result is stated in the more general setting, where our MFG begins at time $t_0 \in [0, T]$, with the initial distribution specified as $\mu(t_0, x) = \eta(x)$. Then, $u^{t_0, \eta}$ and $\mu^{t_0, \eta}$ describe the evolution of the value function and flow of measures starting at time $t_0$; this formalism is necessary for results concerning the master equation, but it is not directly relevant to our setting, so we assume that $t_0 = 0$ throughout.

## D  PICARD ITERATION FOR FORWARD-BACKWARD SYSTEMS

In this section, we describe the precise details of the Picard iteration map, denoted by $\Gamma_g$, that we use as a subprocess in Algo. 1 for sampling from parametrized families of finite-state MFGs. Specifically, we recall the forward-backward MFG system from (2.3):

$$\frac{d}{dt}u^{\eta,\kappa}(t, x) + \bar{H}(x, \mu^{\eta,\kappa}(t), \Delta_x u^{\eta,\kappa}(t, \cdot)) = 0, \qquad (t, x) \in [0, T] \times [d],$$

$$\frac{d}{dt}\mu^{\eta,\kappa}(t, x) = \sum_{y \in [d]} \mu^{\eta,\kappa}(t, y)\gamma_x^*(y, \Delta_y u^{\eta,\kappa}(t, \cdot)), \qquad (t, x) \in [0, T] \times [d],$$

$$\mu^{\eta,\kappa}(0, x) = \eta(x), \qquad x \in [d],$$
$$u^{\eta,\kappa}(T, x) = g_\kappa(x, \mu^{\eta,\kappa}(T)), \qquad x \in [d].$$

To solve this ODE system on the time interval $[0, T]$, we introduce a time discretization with $M$ points and time step $\Delta t := 1/M$, partitioning the interval $[0, T]$ into subintervals $[t_i, t_{i+1}]$ with $t_i = i\Delta t$ for $i = 0, \ldots, M$. Then, for each $i = 0, \ldots, M-1$, the time-discretized system becomes a nonlinear system of equations given by

$$u^{\eta,\kappa}(t_{i+1}, x) - u^{\eta,\kappa}(t_i, x) = -\Delta t \overline{H}(x, \mu^{\eta,\kappa}(t_{i+1}), \Delta_x u^{\eta,\kappa}(t_i, \cdot)), \qquad x \in [d],$$

$$\mu^{\eta,\kappa}(t_{i+1}, x) - \mu^{\eta,\kappa}(t_i, x) = \Delta t \sum_{y \in [d]} \mu^{\eta,\kappa}(t_i, y)\gamma_x^*(y, \Delta_y u^{\eta,\kappa}(t_{i+1}, \cdot)), \qquad x \in [d],$$

$$\mu^{\eta,\kappa}(t_0, x) = \eta(x), \qquad x \in [d],$$
$$u^{\eta,\kappa}(t_M, x) = g_\kappa(x, \mu^{\eta,\kappa}(t_M)), \qquad x \in [d].$$
$$\tag{D.1}$$

Using fixed point iteration, we produce approximations of the value function $\{u^{\eta,\kappa}(t_i, \cdot)\}_{i=0}^M$ and flow of measures of $\{\mu^{\eta,\kappa}(t_i, \cdot)\}_{i=0}^M$, evaluated along the time discretization $t_0, \ldots, t_M$. Now, given

*fixed* $\kappa \in \mathcal{K}$, the output of the Picard iteration map $\Gamma : \mathcal{P}([d]) \to (\mathbb{R}^d)^{M+1}$ is $\Gamma(\eta)_i \approx u^{\eta,\kappa}(t_i, \cdot) \in \mathbb{R}^d$.

For ease of notation, we suppress the dependence of $\mu$ and $u$ on $(\eta, \kappa) \in \mathcal{P}([d]) \times \mathcal{K}$ below, noting that this method solves a *single* MFG from a parametrized family. To begin, we initialize vectors $\mu^{(0)} \in \mathbb{R}^{M+1}$ and $u^{(0)} \in \mathbb{R}^{M+1}$ with $\mu_0^{(0)}(x) = \eta(x)$ and $u_M^{(0)}(x) = g_\kappa(x, \mu_M^{(0)})$ for $x \in [d]$. Then, we alternate between updates to $u$ and $\mu$ via the finite difference equations in (D.1), producing iterates $u^{(k)} \in \mathbb{R}^M$ and $\mu^{(k)} \in \mathbb{R}^k$ in an alternating fashion. If the map $u^{(i)} \mapsto u^{(i+1)}$ is a strict contraction, then a standard argument via the Banach fixed point theorem shows that this iterative procedure will converge the solution $u \in (\mathbb{R}^d)^{M+1}$ to the time-discretized Equation (D.1). Importantly, note that the discretization of the time derivative incurs an error of $\mathcal{O}(\Delta t)$, so we must take $\Delta t$ small in order for fixed point iteration be accurate.

As discussed in (Laurière, 2021, Section 2.3), Picard iteration for such forward-backward systems may sometimes be numerically unstable. If this is the case, we may introduce a sequence of damping parameters $\{\delta^{(k)}\}_{k\in\mathbb{N}}$ and carry out damped updates to one of the updates. For instance, in Laurière (2021), the author includes an auxiliary update $\tilde{\mu}^{(k)}$, with $\mu^{(0)} = \tilde{\mu}^{(0)}$, and updates the forward equation via

$$\tilde{\mu}^{(k+1)} = \delta^{(k)}\tilde{\mu}^{(k)} + (1 - \delta^{(k)})\mu^{(k)}$$

to encourage more stable convergence. Then, the backward equation is updated with $\tilde{\mu}^{(k)}$; the update to the forward equation remains the same. This algorithm, based on (Laurière, 2021, Algorithm 1), is included below. In the numerical examples in Section 5, we do not require damping in order for fixed point iteration to converge quickly and we simply take $\delta^{(k)} = 0$ for all $k \in \mathbb{N}$. However, for more complex MFG systems, damping may be a helpful augmentation of our sampling procedure. Algo. 2 provides a summary of the procedure outlined above.

---

**Algorithm 2** Picard Iteration for Time-Discretized MFGs

---

**Input:** Parameters $(\eta, \kappa) \in \mathcal{P}([d]) \times \mathcal{K}$, number of time steps $M \in \mathbb{N}$, tolerance $\varepsilon > 0$, damping schedule $\{\delta^{(k)}\}_{k\in\mathbb{N}}$, initializations $u_0, \mu_0 \in (\mathbb{R}^d)^{M+1}$

1: $u^{(0)} \leftarrow u_0$
2: $\mu^{(0)} \leftarrow \mu_0$
3: $\tilde{\mu}^{(0)} \leftarrow \mu_0$
4: $k \leftarrow 0$
5: **while** $\|u^{(k+1)} - u^{(k)}\|_2 \geq \varepsilon$ or $\|\mu^{(k+1)} - \mu^{(k)}\|_2 \geq \varepsilon$ **do**
6:     Solve the discretized backward equation in Equation (D.1) for $u^{(k+1)}$, with input $\tilde{\mu}^{(k)}$.
7:     Solve the discretized foward equation in Equation (D.1) for $\mu^{(k+1)}$, with input $u^{(k+1)}$.
8:     $\tilde{\mu}^{(k+1)} \leftarrow \delta^{(k)}\tilde{\mu}^{(k)} + (1 - \delta^{(k)})\mu^{(k)}$
9:     $k \leftarrow k + 1$
10: **end while**
11: **return** $u^{(k)}$

---

# E  TECHNICAL PROOFS

In this section, we present technical lemmata and proofs for our claims about the regularity of flow maps for parametrized families of MFGs. First, we recall some useful notation. For any compact set $K \subseteq \mathbb{R}^d$ and a function $\phi : K \to \mathbb{R}$, we define

$$\|\phi\|_\infty := \sup_{x \in K} |\phi(x)|.$$

All functions such that $\|\phi\|_\infty < \infty$ form the Banach space $\mathcal{C}^0(K)$. For instance, for functions such as $u : [0, T[\times[d] \to \mathbb{R}$, we take

$$\|u\|_\infty = \sup_{t \in T} \max_{x \in [d]} |u(t, x)|.$$

We also occasionally refer to the spaces $\mathcal{C}^{0,1}(K)$, consisting of all Lipschitz functions on $K$, and $\mathcal{C}^{0,1}(K)$, consisting of all continuously differentiable functions on $K$ with Lipschitz derivatives. For

functions on the $d$-dimensional probability simplex $\mathcal{P}([d])$, we only allow directional derivatives along the directions $e_y - e_x$, where $e_x, e_y$ are standard basis vectors in $\mathbb{R}^d$.

### E.1 PROOFS OF REGULARITY RESULT

For our regularity results, (Cecchin & Pelino, 2019, Proposition 5) provides a very useful starting point. Importantly, the authors of Cecchin & Pelino (2019) work under Assumptions 2.1. It remains to incorporate the added effect of a changing terminal condition, restricted to a parametrized set of functions under Assumption 2.2, into their results.

First, we define $\tilde{\Phi}(t, \eta) := U(t, \cdot, \mu^\eta(t)) = u(t, \cdot)$. due to the consistency relation in Proposition C.1 in Appendix C, where $U$ is the solution to the master equation. Now, Proposition C.1 also provides that $U(\cdot, x, \cdot) \in \mathcal{C}^{1,1}([0, T] \times \mathcal{P}([d]))$ for every $x \in [d]$ under our assumptions, which directly implies the following:

**Lemma E.1.** *Under Assumption 2.1, The flow map* $\tilde{\Phi} : [0, T] \times \mathcal{P}([d]) \to \mathbb{R}^d$, *given by* $\tilde{\Phi}(t, \eta) = u^\eta(t, \cdot)$, *satisfies* $\Phi \in \mathcal{C}^{1,1}([0, T] \times \mathcal{P}([d]); \mathbb{R}^d)$.

Including Assumption 2.2, on top of Assumptions 2.1, we show that the flow map

$$\Phi : [0, T] \times \mathcal{P}([d]) \times \mathcal{K} \to \mathbb{R}^d, \quad \Phi(t, \eta, \kappa) = u^{\eta,\kappa}(t, \cdot)$$

is Lipschitz in all three arguments. Above, recall that the notation $u^{\eta,\kappa}$ denotes the value function that solves the MFG system, with initial distribution $\eta \in \mathcal{P}([d])$ and terminal cost $g_\kappa$, where $\kappa \in \mathcal{K}$. In turn, Lipschitz regularity of the flow map on the compact set $[0, T] \times \mathcal{P}([d]) \times \mathcal{K}$, recalling that Assumption 2.2 requires that $\mathcal{K}$ is compact, is sufficient to invoke the approximation guarantees provided in Jiao et al. (2023). We begin with a stability estimate for the parametrized family of MFG systems obeying Assumption 2.2.

**Lemma E.2.** *Let* $(u_1, \mu_1)$ *and* $(u_2, \mu_2)$ *solve the MFG system in* (2.3) *with data* $(\eta_1, g_{\kappa_1})$ *and* $(\eta_2, g_{\kappa_2})$ *respectively, with* $\eta_1, \eta_2 \in \mathcal{P}([d])$ *and* $\kappa_1, \kappa_2 \in \mathcal{K} \subset \mathbb{R}^k$. *If Assumptions 2.1–2.2 hold, then there exists a constant* $C > 0$ *such that*

$$\sup_{t \in [0,T]} \max_{x \in [d]} |u_1(t, x) - u_2(t, x)| \le C(|\kappa_1 - \kappa_2| + \|\mu_1 - \mu_2\|_\infty). \tag{E.1}$$

*Proof.* We proceed as in Cecchin & Pelino (2019), taking $u := u_1 - u_2$ and $\mu = \mu_1 - \mu_2$. The pair $(u, \mu)$ then solves the system

$$\frac{d}{dt}u(t, x) + \bar{H}(x, \mu_1(t), \Delta_x u_1(t, \cdot)) - \bar{H}(x, \mu_2(t), \Delta_x u_2(t, \cdot)) = 0, \qquad (t, x) \in [0, T] \times [d],$$

$$\frac{d}{dt}\mu(t, x) = \sum_{y \in [d]} [\mu_1(t, y)\gamma_x^*(y, \Delta_y u_1(t, \cdot)) - \mu_2(t, y)\gamma_x^*(y, \Delta_y u_2(t, \cdot))], \qquad (t, x) \in [0, T] \times [d],$$

$$\mu(0, x) = \eta_1(x) - \eta_2(x), \qquad x \in [d],$$

$$u(T, x) = g_{\kappa_1}(x, \mu_1(T)) - g_{\kappa_2}(x, \mu_2(T)), \qquad x \in [d]. \tag{E.2}$$

To begin, we integrate the backward-in-time HJB equation in (E.2) over the interval $[t, T]$, where $t \in [0, T]$ to obtain

$$u(t, x) = g_{\kappa_1}(x, \mu_1(T)) - g_{\kappa_2}(x, \mu_2(T)) + \int_t^T \left[ \bar{H}(x, \mu_1(s), \Delta_x u_1(s, \cdot)) - \bar{H}(x, \mu_2(s), \Delta_x u_2(s, \cdot)) \right] ds$$

Observe that

$$|g_{\kappa_1}(x, \mu_1(T)) - g_{\kappa_2}(x, \mu_2(T))| = |g_{\kappa_1}(x, \mu_1(T)) - g_{\kappa_2}(x, \mu_1(T)) + g_{\kappa_2}(x, \mu_1(T)) - g_{\kappa_2}(x, \mu_2(T))|$$
$$\le C(|\kappa_1 - \kappa_2| + |\mu_1(T) - \mu_2(T)|)$$
$$\le C(|\kappa_1 - \kappa_2| + \|\mu_1 - \mu_2\|_\infty).$$

leveraging both Assumption 2.2 and the fact that $g_\kappa(x, \cdot) \in \mathcal{C}^1(\mathcal{P}([d]))$ so that $g_{\kappa_2}$ is Lipschitz in its second input. Now, recall that

$$\bar{H}(x, \eta, b) = H(x, b) + F(x, \eta),$$

with $H$ Lipschitz in $b$ and $F$ Lipschitz in $\eta$ under Assumptions 2.1 and 2.1. Consequently, we have that

$$|\bar{H}(x, \mu_1(s), \Delta_x u_1(s, \cdot)) - \bar{H}(x, \mu_2(s), \Delta_x u_2(s, \cdot))| \le C \left(|\mu_1(s) - \mu_2(s)| + |\Delta_x u_1(s, \cdot) - \Delta_x u_2(s, \cdot)|\right)$$
$$\le C(|\mu_1(s) - \mu_2(s)| + \max_{x \in [d]} |u(s, x)|),$$

recognizing that

$$|\Delta_x u_1(s, \cdot) - \Delta_x u_2(s, \cdot)|^2 = |\Delta_x u(s, \cdot)|^2 = \sum_{y \in [d]} (u(s, y) - u(s, x))^2 \le 3d \max_{x \in [d]} |u(s, x)|^2.$$

Taking absolute values and the maximum over $x \in [d]$ of the integrated HJB equation, we are left with

$$\max_{x \in [d]} |u(t, x)| \le C(|\kappa_1 - \kappa_2| + \|\mu_1 - \mu_2\|_\infty) + C \int_t^T |\mu_1(s) - \mu_2(s)| ds + C \int_t^T \max_{x \in [d]} |u(s, x)| ds$$

$$\le C(|\kappa_1 - \kappa_2| + \|\mu_1 - \mu_2\|_\infty) + C \int_t^T \max_{x \in [d]} |u(s, x)| ds.$$

Applying a reversed version of Gronwall's inequality, we obtain

$$\max_{x \in [d]} |u(t, x)| \le C(|\kappa_1 - \kappa_2| + \|\mu_1 - \mu_2\|_\infty)$$

for all $t \in [0, T]$ so that

$$\|u_1 - u_2\|_\infty \le C(|\kappa_1 - \kappa_2| + \|\mu_1 - \mu_2\|_\infty),$$

taking the supremum over $t \in [0, T]$. □

Next, we require an estimate on the difference $\|\mu_1 - \mu_2\|_\infty$; this time, the argument from Cecchin & Pelino (2019) applies without modification.

**Lemma E.3.** *Under the same assumptions as in Lemma E.2, the difference in measures satisfies*

$$\|\mu_1 - \mu_2\|_\infty \le C|\eta_1 - \eta_2| + C \int_0^T \sqrt{\sum_{x \in [d]} |\Delta_x (u_1 - u_2)(s, \cdot)|^2 \mu_1(s, x)} ds.$$

*Proof.* This estimate follows by integrating the (forward) Kolmogorov equation for $\mu$ from (E.2); see (Cecchin & Pelino, 2019, Proposition 5) for details, which carry over verbatim to our setting. □

Equipped with both of the previous lemmata, we proceed to bound $\|u\|_\infty$ and $\|\mu\|_\infty$ in terms of the initial-terminal data $(\eta_1, \kappa_1)$ and $(\eta_2, \kappa_2)$.

**Lemma E.4.** *Let $(u_1, \mu_1)$ and $(u_2, \mu_2)$ solve the MFG system in (2.3) with data $(\eta_1, g_{\kappa_1})$ and $(\eta_2, g_{\kappa_2})$ respectively, with $\eta_1, \eta_2 \in \mathcal{P}([d])$ and $\kappa_1, \kappa_2 \in \mathcal{K} \subset \mathbb{R}^k$. If Assumptions 2.1–2.2 hold, then there exists a constant $C > 0$ such that*

$$\|\mu_1(t, x) - \mu_2(t, x)\|_\infty \le C(|\eta_1 - \eta_2| + |\kappa_1 - \kappa_2|), \tag{E.3}$$
$$\|u_1 - u_2\|_\infty \le C(|\eta_1 - \eta_2| + |\kappa_1 - \kappa_2|) \tag{E.4}$$

As a direct corollary, we can extend this stability result to obtain Lipschitz continuity of the flow map $\Phi : [0, T] \times \mathcal{P}([d]) \times \mathcal{K} \to \mathbb{R}^d$.

*Proof of Lemma E.4.* Taking $\phi(t) = \langle u(t, \cdot), \mu(t, \cdot) \rangle$, we see that

$$\phi'(t) = \sum_{x \in [d]} u(t, x) \frac{d\mu}{dt}(t, x) + \sum_{x \in [d]} \frac{du}{dt}(t, x)\mu(t, x)$$

$$= \sum_{x \in [d]} \sum_{y \in [d]} [\mu_1(t, y)\gamma_x^*(y, \Delta_y u_1(t, \cdot)) - \mu_2(t, y)\gamma_x^*(y, \Delta_y u_2(t, \cdot))](u_1(t, x) - u_2(t, x))$$

$$+ \sum_{x \in [d]} [\bar{H}(x, \mu_2(t), \Delta_x u_1(t, \cdot)) - \bar{H}(x, \mu_1(t), \Delta_x u_2(t, \cdot))](\mu_1(t, x) - \mu_2(t, x)).$$

Integrating over the interval $[0, T]$, we obtain

$$\phi(T) - \phi(0) = \int_0^T \left[ \sum_{x \in [d]} \sum_{y \in [d]} [\mu_1(t, y)\gamma_x^*(y, \Delta_y u_1(t, \cdot)) - \mu_2(t, y)\gamma_x^*(y, \Delta_y u_2(t, \cdot))](u_1(t, x) - u_2(t, x)) \right] dt$$

$$+ \int_0^t \left[ \sum_{x \in [d]} [\bar{H}(x, \mu_2(t), \Delta_x u_2(t, \cdot)) - \bar{H}(x, \mu_1(t), \Delta_x u_1(t, \cdot))](\mu_1(t, x) - \mu_2(t, x)) \right] dt.$$

In the first integral, we observe that under Assumption 2.1, we have that

$$\sum_{x \in [d]} \gamma_x^*(y, \cdot) = 0.$$

As a result, we can interchange the order of summation to obtain

$$\sum_{x \in [d]} \sum_{y \in [d]} [\mu_1(t, y)\gamma_x^*(y, \Delta_y u_1(t, \cdot)) - \mu_2(t, y)\gamma_x^*(y, \Delta_y u_2(t, \cdot))](u_1(t, x) - u_2(t, x))$$

$$= \sum_{y \in [d]} \sum_{x \in [d]} [\mu_1(t, y)\gamma_x^*(y, \Delta_y u_1(t, \cdot)) - \mu_2(t, y)\gamma_x^*(y, \Delta_y u_2(t, \cdot))](u_1(t, x) - u_1(t, y) + u_2(t, y) - u_2(t, x))$$

$$= \sum_{y \in [d]} \sum_{x \in [d]} [\mu_1(t, y)\gamma_x^*(y, \Delta_y u_1(t, \cdot)) - \mu_2(t, y)\gamma_x^*(y, \Delta_y u_2(t, \cdot))]\Delta_y u(t, x)$$

$$= \sum_{x \in [d]} \sum_{y \in [d]} [\mu_1(t, x)\gamma_y^*(x, \Delta_x u_1(t, \cdot)) - \mu_2(t, x)\gamma_y^*(x, \Delta_y u_2(t, \cdot))]\Delta_x u(t, y)$$

$$= \sum_{x \in [d]} \Delta_x u \cdot [\mu_1(t, x)\gamma^*(x, \Delta_x u_1(t, \cdot)) - \mu_2(t, x)\gamma^*(x, \Delta_x u_2(t, \cdot))],$$

switching the role of $x$ and $y$ in the fourth line for notational consistency below. With this, we see that

$$\sum_{x \in [d]} (g_{\kappa_1}(x, \mu_1(T)) - g_{\kappa_2}(x, \mu_2(T)))[\mu_1(T, x) - \mu_2(T, x)]$$

$$= \sum_{x \in [d]} (u_1(0, x) - u_2(0, x))[\eta_1(x) - \eta_2(x)]$$

$$+ \int_0^T \left[ \sum_{x \in [d]} [\bar{H}(x, \mu_2(t), \Delta_x u_2(t, \cdot)) - \bar{H}(x, \mu_1(t), \Delta_x u_1(t, \cdot))](\mu_1(t, x) - \mu_2(t, x)) \right] dt$$

$$+ \int_0^T \left[ \sum_{x \in [d]} \Delta_x u \cdot (\mu_1(t, x)\gamma^*(x, \Delta_x u_1(t, \cdot)) - \mu_2(t, x)\gamma^*(x, \Delta_x u_2(t, \cdot))) \right] dt.$$

$$(E.5)$$

At this point, we note that the lefthand side of the above equality can be decomposed as

$$\sum_{x \in [d]} (g_{\kappa_1}(x, \mu_1(T)) - g_{\kappa_2}(x, \mu_2(T)))[\mu_1(T, x) - \mu_2(T, x)]$$

$$= \sum_{x \in [d]} (g_{\kappa_1}(x, \mu_1(T)) - g_{\kappa_1}(x, \mu_2(T)) + g_{\kappa_1}(x, \mu_2(T)) - g_{\kappa_2}(x, \mu_2(T)))[\mu_1(T, x) - \mu_2(T, x)]$$

$$\geq \sum_{x \in [d]} (g_{\kappa_1}(x, \mu_2(T)) - g_{\kappa_2}(x, \mu_2(T)))[\mu_1(T, x) - \mu_2(T, x)],$$

invoking the fact that $g_{\kappa_1}$ is Lasry–Lions monotone; see Assumption 2.1. Now, we use Assumption 2.2 to bound

$$\left| \sum_{x \in [d]} (g_{\kappa_1}(x, \mu_2(T)) - g_{\kappa_2}(x, \mu_2(T)))[\mu_1(T, x) - \mu_2(T, x)] \right|$$

$$\leq \sum_{x \in [d]} |g_{\kappa_1}(x, \mu_2(T)) - g_{\kappa_2}(x, \mu_2(T))||\mu_1(T, x) - \mu_2(T, x)|$$

$$\leq C|\kappa_1 - \kappa_2|\|\mu_1 - \mu_2\|_\infty$$

absorbing additional constants into $C$ as necessary (e.g., $C$ absorbs a factor of $d$ in the final line). In summary,

$$\sum_{x \in [d]} (g_{\kappa_1}(x, \mu_1(T)) - g_{\kappa_2}(x, \mu_2(T)))[\mu_1(T, x) - \mu_2(T, x)] \geq -C(|\kappa_1 - \kappa_2|^2 + \|\mu_1 - \mu_2\|_\infty^2).$$

On the other hand, observe that

$$\sum_{x \in [d]} \left[ \bar{H}(x, \mu_2(t), \Delta_x u_2(t, \cdot)) - \bar{H}(x, \mu_1(t), \Delta_x u_1(t, \cdot))](\mu_1(t, x) - \mu_2(t, x)) \right]$$

$$= \sum_{x \in [d]} [H(x, \Delta_x u_2(t, \cdot)) - H(x, \Delta_x u_1(t, \cdot))(\mu_1(t, x) - \mu_2(t, x))]$$

$$- \sum_{x \in [d]} (F(x, \mu_1(t)) - F(x, \mu_2(t)))(\mu_1(t, x) - \mu_2(t, x))$$

$$\leq \sum_{x \in [d]} [H(x, \Delta_x u_2(t, \cdot)) - H(x, \Delta_x u_1(t, \cdot))(\mu_1(t, x) - \mu_2(t, x))],$$

recalling that $F$ also satisfies the Lasry–Lions monotonicity assumption from Assumption 2.1. Now, (Gomes et al., 2013, Proposition 1) implies that

$$\gamma^*(x, p) = D_p H(x, p). \tag{E.6}$$

From this, we have that

$$\gamma^*(x, \Delta_x u_i(t, \cdot)) = D_p H(x, \Delta_x u_i(t, \cdot)), \quad i = 1, 2,$$

allowing us to write

$$H(x, \Delta_x u_2(t, \cdot)) - H(x, \Delta_x u_1(t, \cdot)) + \Delta_x u \cdot \gamma^*(x, \Delta_x u_1(t, \cdot))$$

$$= H(x, \Delta_x u_2(t, \cdot)) - [H(x, \Delta_x u_1(t, \cdot)) + (\Delta_x u_2 - \Delta_x u_1) \cdot D_p H(x, \Delta_x u_1(t, \cdot)]$$

$$\leq -C_{2,H}|\Delta_x u|^2$$

by Assumption (2.1). Namely, the Hessian $D_{pp}^2 H(x, p)$ exists and satisfies the bound $D_{pp}^2 H(x, p) \leq -C_{2,H}$ for some constant $C_{2,H} \geq 0$ under our strict concavity assumption. By the same reasoning, we observe that

$$H(x, \Delta_x u_1(t, \cdot)) - H(x, \Delta_x u_2(t, \cdot)) - \Delta_x u \cdot \gamma^*(x, \Delta_x u_2(t, \cdot))$$

$$= H(x, \Delta_x u_1(t, \cdot)) - [H(x, \Delta_x u_2(t, \cdot)) + (\Delta_x u_1 - \Delta_x u_2) \cdot D_p H(x, \Delta_x u_2(t, \cdot)]$$

$$\leq -C_{2,H}|\Delta_x u|^2.$$

Thus, returning to (E.5), we have that

$$-C(|\kappa_1 - \kappa_2|\|\mu_1 - \mu_2\|_\infty) \leq \sum_{x \in [d]} (u_1(0, x) - u_2(0, x))[\eta_1(x) - \eta_2(x)]$$

$$- C \int_0^T \sum_{x \in [d]} |\Delta_x u(s, \cdot)|^2 (\mu_1(s, x) + \mu_2(s, x)) ds$$

Upon rearrangement, and an application of the Cauchy–Schwarz inequality to the first term on the righthand side of the above inequality, it follows that

$$\int_0^T \sum_{x \in [d]} |\Delta_x u(s, \cdot)|^2 (\mu_1(s, x) + \mu_2(s, x)) ds \leq C(\|u\|_\infty |\eta_1 - \eta_2| + |\kappa_1 - \kappa_2| \|\mu\|_\infty)$$

for some constant $C > 0$. Now, invoking Lemma E.3, the Cauchy–Schwarz inequality, and the fact that $\mu_2(s, x) \geq 0$ for all $s \in [0, T]$ and $x \in [d]$, we have that

$$\|\mu\|_\infty \leq C|\eta_1 - \eta_2| + C \int_0^T \sqrt{\sum_{x \in [d]} |\Delta_x u(s, \cdot)|^2 \mu_1(s, x)} ds$$

$$\leq C|\eta_1 - \eta_2| + C \sqrt{\int_0^T \sum_{x \in [d]} |\Delta_x u(s, \cdot)|^2 \mu_1(s, x) ds}$$

$$\leq C(|\eta_1 - \eta_2| + \sqrt{\|u\|_\infty |\eta_1 - \eta_2| + |\kappa_1 - \kappa_2| \|\mu\|_\infty})$$

$$\leq C(|\eta_1 - \eta_2| + \|u\|_\infty^{1/2} |\eta_1 - \eta_2|^{1/2} + |\kappa_1 - \kappa_2|^{1/2} \|\mu\|_\infty^{1/2}),$$

recalling that $\sqrt{a} + \sqrt{b} \geq \sqrt{a + b}$ for any $a, b \geq 0$. Now, recall that for any $a, b \geq 0$ and $\varepsilon > 0$, we also have that

$$ab \leq \varepsilon a^2 + \frac{1}{4\varepsilon} b^2.$$

Applying this inequality once with $\varepsilon = \frac{1}{2C}$, we see that

$$\|\mu\|_\infty \leq C(|\eta_1 - \eta_2| + |\kappa_1 - \kappa_2| + \|u\|_\infty^{1/2} |\eta_1 - \eta_2|^{1/2}) + \frac{1}{2} \|\mu\|_\infty$$

taking $C > 0$ larger if necessary. Applying the same inequality again with $\varepsilon = \frac{1}{4C^2}$ and rearranging, it follows that

$$\|\mu\|_\infty \leq C(|\eta_1 - \eta_2| + |\kappa_1 - \kappa_2|) + \frac{1}{2C} \|u\|_\infty. \tag{E.7}$$

Plugging this into the result of Lemma E.2 and rearranging yields

$$\|u\|_\infty \leq C(|\eta_1 - \eta_2| + |\kappa_1 - \kappa_2|), \tag{E.8}$$

and plugging (E.8) into (E.7) results in

$$\|\mu\|_\infty \leq C(|\eta_1 - \eta_2| + |\kappa_1 - \kappa_2|)$$

as claimed. $\square$

To conclude, we can present the proof of our main theorem, which follows almost immediately from the preceding results.

*Proof of Theorem 4.1.* Observe that we can write

$$|\Phi(t, \eta_1, \kappa_1) - \Phi(s, \eta_2, \kappa_2)| = |\Phi(t, \eta_1, \kappa_1) - \Phi(s, \eta_1, \kappa_1) + \Phi(s, \eta_1, \kappa_1) - \Phi(s, \eta_2, \kappa_2)|$$

$$\leq |\Phi(t, \eta_1, \kappa_1) - \Phi(s, \eta_1, \kappa_1)| + |\Phi(s, \eta_1, \kappa_1) - \Phi(s, \eta_2, \kappa_2)|$$

$$\leq C(|t - s| + |\eta_1 - \eta_2| + |\kappa_1 - \kappa_2|),$$

invoking Lemma E.1 to bound the first term and Lemma E.4 to bound the second term. $\square$

**Remark E.1.** *Although Lanthaler & Stuart (2025) reference the approximation guarantee from (Yarotsky, 2017, Theorem 1) to show that their HJ-Net method evades the curse of parametric complexity, most existing guarantees on the generalization performance of ReLU neural networks require bounds on the weights of the neural network rather than the size of the network. The well-known result from Yarotsky (2017), however, only provides width and depth bounds on ReLU networks approximating a function with prescribed regularity. To this end, we pursue an alternative approach for obtaining approximation and generalization guarantees, based on the recent results of Jiao et al. (2023).*

## E.2 PROOFS OF APPROXIMATION AND GENERALIZATION GUARANTEES

We conclude with proofs of Corollary 4.3 and Corollary 4.5, our approximation and generalization results respectively. Both follow almost directly from the corresponding results in Jiao et al. (2023), in Proposition 4.2 and Proposition 4.4 respectively, but we include the necessary rescaling arguments here for the sake of completeness.

*Proof of Corollary 4.3.* First, to extend from the setting of scalar regression, as is the case in Proposition 4.2, to vector-valued regression, we note that if a scalar Lipschitz function can be uniformly approximated up to an error $\varepsilon > 0$ by a network with weight bound $K$ and width $W$, then an $\mathbb{R}^d$ valued Lipschitz function $\Phi$ can be uniformly approximated by a network with weight bound $K$ and width $dW$. Indeed, we can simply approximate each coordinate of $\Phi$ by a network of width $W$ and stack the resulting networks to obtain the desired approximator, which will have width $dW$.

Now, by Theorem 4.1, the flow map $\Phi : [0, T] \times \mathcal{P}([d]) \times \mathcal{K} \to \mathbb{R}^d$ belongs to $\mathcal{C}^{0,1}([0, T] \times \mathcal{P}([d]) \times \mathcal{K})$. From this, we can apply Proposition 4.2 directly upon scaling the domain $[0, T] \times \mathcal{P}([d]) \times \mathcal{K}$ to lie entirely within the $(d + k + 1)$-dimensional unit cube.

To carry out this scaling, we embed $\mathcal{P}([d]) \hookrightarrow [0, 1]^d$, scale $\mathcal{K}$ to lie in the set $[0, 1]^k$, and scale the interval $[0, T]$ to lie in the interval $[0, 1]$. The natural embedding $\mathcal{P}([d]) \hookrightarrow [0, 1]^d$ is simply given by viewing

$$\mathcal{P}([d]) = \left\{ \eta \in \mathbb{R}^d : \sum_{i=1}^{d} \eta_i = 1, \quad \eta_i \geq 0 \text{ for all } i = 1, \ldots, d \right\}.$$

This rescaling may incur constants that depend on the diameter of $\mathcal{K}$, denoted by $\mathrm{diam}(\mathcal{K})$, and the final time $T$. Importantly, it is always possible for finite $T > 0$ and compact $\mathcal{K} \subset \mathbb{R}^k$. The result then follows upon applying Proposition 4.2, replacing $d$ with $d + k + 1$ therein. As noted above, the universal constants $c, C > 0$ obtained in Proposition 4.2 must also be replaced by constants $c(\mathrm{diam}(\mathcal{K}), T), C(\mathrm{diam}(\mathcal{K}), T) > 0$ that depends on $\mathcal{K}$ and $T$. □

*Proof of Corollary 4.5.* This follows directly from Proposition 4.4 upon carrying out the same rescaling argument as in the previous proof, again replacing $d$ with $d + k + 1$ in the statement of the result. Again, we note that the universal constant $c > 0$ from Proposition 4.4 must be replaced by a constant $\tilde{C}(\mathrm{diam}(\mathcal{K}), T) > 0$ that can depend on $\mathcal{K}$ and $T$. □

## F CONNECTION TO HAMILTONIAN FLOW

In this appendix, we expand upon the similarity between the MFG system and the characteristic ODEs that Lanthaler & Stuart (2025) utilize to obtain parameter-efficient operator learning for first-order HJB equations. Consider an arbitrary first-order HJB equation on a bounded domain $\Omega \subseteq \mathbb{R}^d$, with Hamiltonian $H : \mathbb{R}^d \times \mathbb{R}^d \to \mathbb{R}$:

$$\begin{cases} \partial_t u + H(q, \nabla_q u) = 0 & (x, t) \in \Omega \times (0, T], \\ u(x, 0) = u_0(x) & x \in \Omega, \end{cases} \tag{F.1}$$

Instead of attempting to learn the operator that maps the initial data $u_0 \in C^r(\Omega)$ to $u \in C^r(\Omega \times [0, T])$, for instance, Lanthaler & Stuart (2025) construct a scheme they label HJ-Net with the aim of learning the Hamiltonian flow (i.e., the characteristics of the HJB equation), which satisfies the ODE system

$$\begin{cases} \dot{q} = \nabla_p H(q, p) & q(0) = q_0, , \\ \dot{p} = -\nabla_q H(q, p) & p(0) = p_0, \\ \dot{z} = \mathcal{L}(q, p) & z(0) = z_0. \end{cases} \tag{F.2}$$

Then, the flow map $\Psi_t : \Omega \times \mathbb{R}^d \times \mathbb{R} \to \Omega \times \mathbb{R}^d \times \mathbb{R}$, given by $(q_0, p_0, z_0) \mapsto (q(t), p(t), z(t))$ is such that $z(t) = u(q(t), t)$ and $p(t) = \nabla_q u(q(t), t)$ along the characteristics $(q(t), t)$. By learning the flow map $\Psi_t$, instead of the operator $u_0 \mapsto u$, and reconstructing the solution $u$ from the characteristics, (Lanthaler & Stuart, 2025, Theorem 5.1) shows that the HJ-Net approach can beat the so-called

curse of parametric complexity, enabling parameter-efficient operator learning for HJB equations. Observe, nonetheless, that there is a subtle but important difference between the Hamiltonian flow and the MFG system: the former is independent of the initial condition $u_0$ of Equation (F.1), while the latter *depends* explicitly on the terminal condition $g_\kappa$. In the setting of Lanthaler & Stuart (2025), this enables parameter-efficient operator learning over initial conditions belonging to an infinite-dimensional Banach space, as the Hamiltonian flow map remains approximable by neural networks of bounded width and depth regardless of the space to which the initial conditions belong. Conversely, for finite-state MFGs, we must limit ourselves to parametrized terminal costs due to the dependence of the MFG system on the terminal cost. Indeed, the technical results in both our work and in Lanthaler & Stuart (2025) rely upon reducing to a flow map between subsets of finite-dimensional Euclidean spaces, which is *not* the case if we allow terminal costs to belong to an infinite-dimensional Banach space.

## G    ADDITIONAL NUMERICAL EXPERIMENTS

We provide a comprehensive suite of additional numerical experiments for both the cybersecurity model and the quadratic model. As alluded to earlier (see also Appendix A), Fig. 7 demonstrates the improvement in accuracy and reduced variance over trials that comes with a more powerful neural network architecture. In particular, we replicate the $d = 10$ results using a ResNet architecture, with layer normalization, skip connections between all layers, a dropout rate of $p = 0.05$. Moreover, the ResNet's first and layer layer have width $W_1 = 128$ while the middle two hidden layers have width $W_2 = 64$. We find that this "bottleneck" helps promote training stability, and Fig. 7 demonstrates the effect that this architecture choice has on accuracy and variance (the latter is illustrated by smaller standard deviations about the mean of the five trials).

Next, Figure 8 and Figure 10 provide additional evidence for the accuracy of our method on the cybersecurity model. Figure 9 displays the errors obtained when learning the flow map for the cybersecurity model, in the same vein as Figure 5 above. Similarly, Figures 12–15 illustrate a variety of random tests for the quadratic model in dimensions $d = 3, 4, 5$, and 10. In Table 3, we present statistics for the models used to produce Figures 12–15 (as well as Figure 4), including average test losses on the held-out test set at the end of training and average training times. Figure 16 is the analogue of Figure 3 for the quadratic model in various dimensions, illustrating that by solving the KFP equation with the learned value function, we can accurately recover the flow of measures for the quadratic model as well.

Finally, we include a handful of figures that learn an operator on a *fixed* time discretization. Specifically, suppose that we discretize the time interval $[0, T]$ with $M$ time, yielding times $t_j = jT/M$ for $j = 0, \ldots, M$. Given a pair $(\eta_i, \kappa_i) \in \mathcal{P}([d]) \times \mathcal{K}$, one may instead attempt to learn the augmented flow map subordinate to the discretization, given by $\tilde{\Phi} : \mathcal{P}([d]) \times \mathcal{K} \to (\mathbb{R}^d)^{M+1}$

$$\tilde{\Phi}(\eta_i, \kappa_i) \mapsto (u^{\eta_i, \kappa_i}(t_j))_{j=0}^M.$$

In practice, this map can be learned using a slight modification of Algo. 1, where the sampling step simply takes in a pair $\tilde{x}_i := (\eta_i, \kappa_i) \in \mathcal{P}([d]) \times \mathcal{K}$ and outputs the entire trajectory that Picard iteration produces as a label, given by $\tilde{y}_i := \Gamma_{g_{\kappa_i}}(\eta_i)$. Then, the pairs $\{(\tilde{x}_i, \tilde{y}_i)\}_{i=1}^n$ become our augmented training data, and we can proceed from Line 7 of Algo. 1 verbatim. Note that the augmented flow map $\tilde{\Phi}$ is less versatile than the flow map $\Phi$ from Section 2.2, in the sense that $\Phi$ can be evaluated at *any* time $t \in [0, T]$, while $\tilde{\Phi}$ can only be evaluated along the given time discretization. However, given $M$ sufficiently large, learning the map $\tilde{\Phi}$ to high precision still yields a useful estimate of the MFG equilibrium, so this modified method may still be of interest.

In Fig. 11, we present an example of the learned map value functions for the cybersecurity model, using the augmented procedure for a fixed time discretization with $M = 50$ points. In Figures 17–21, we provide similar experiments for the quadratic model in dimensions $d = 3, 4, 5, 10, 20$ respectively. Interestingly, the quality of the approximation and optimization stability does not appear to degrade as quickly with dimension, and using a discretization with $M = 10$ points, we are able to learn augmented flow maps to very high precision up to dimension $d = 20$.

Table 3: Statistics for high-dimensional quadratic model experiments. Test losses and training times are averaged over 5 trials, and all networks had depth $L = 4$. The test losses are evaluated using smooth $L^1$ loss, summed over the test set.

| Dimension $d$ | Average Test Loss | Average Training Time (s) | Training Samples | Epochs | Width |
|---|---|---|---|---|---|
| 3 | 0.000831 | 233.42 | 4000 | 2000 | 64 |
| 4 | 0.00200 | 219.68 | 4000 | 2000 | 64 |
| 5 | 0.00527 | 220.24 | 4000 | 2000 | 64 |
| 10 | 0.0208 | 374.10 | 10000 | 500 | 128 |

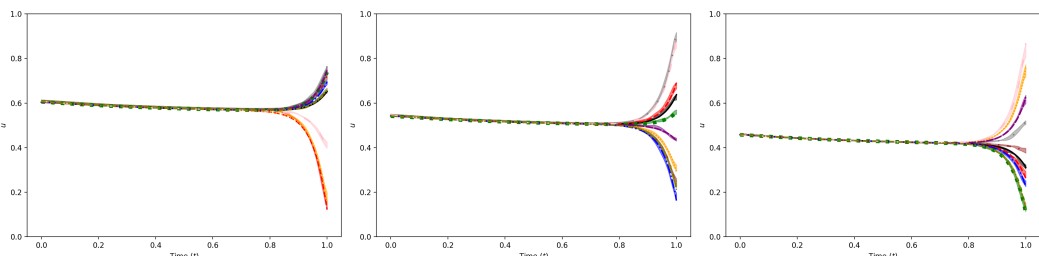

Figure 7: Learned value functions in the same setting as Fig. 20, using a ResNet architecture with dropout, layer normalization, and an hidden layer width of 64.

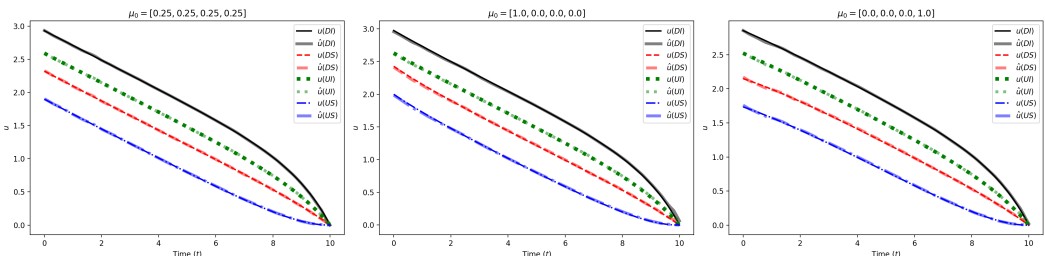

Figure 8: Learned value functions, denoted by $\widehat{u}$, for $\kappa = 0$ and initial distribution $\mu_1 = [0.25, 0.25, 0.25, 0.25]$, $\mu_2 = [1, 0, 0, 0]$, and $\mu_3 = [0, 0, 0, 1]$, respectively. In particular, our method can still perform accurately in the event that the parametrization of the underlying MFG is fixed and only the initial distribution varies.

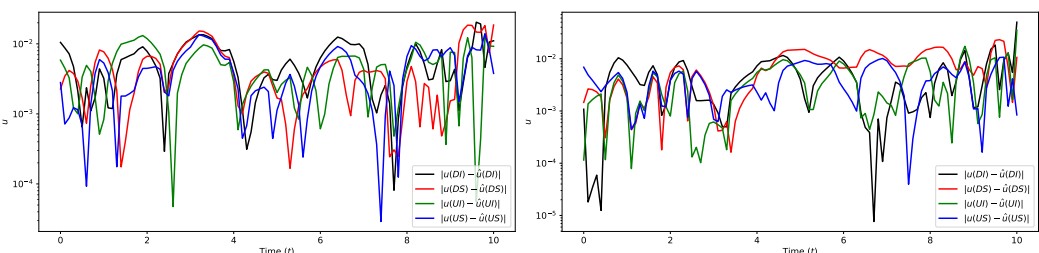

Figure 9: Error, measured as absolute difference across all times between true and learned value function, on two randomly generated instances of the cybersecurity model.

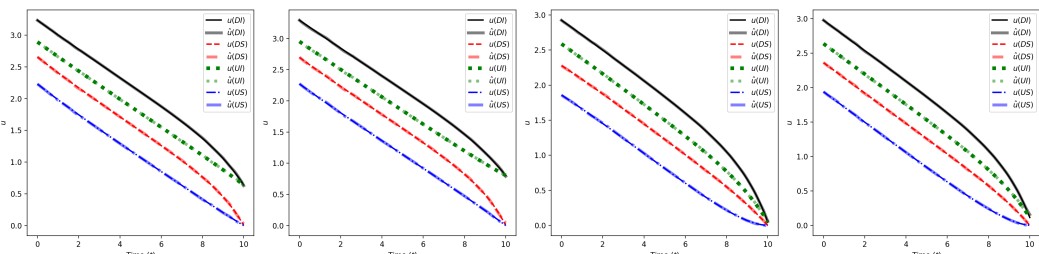

Figure 10: Learned value function, denoted by $\widehat{u}$, approximating time-parametrized flow map $\Phi$, for four random initial distributions and $\kappa \in [0, 1]$.

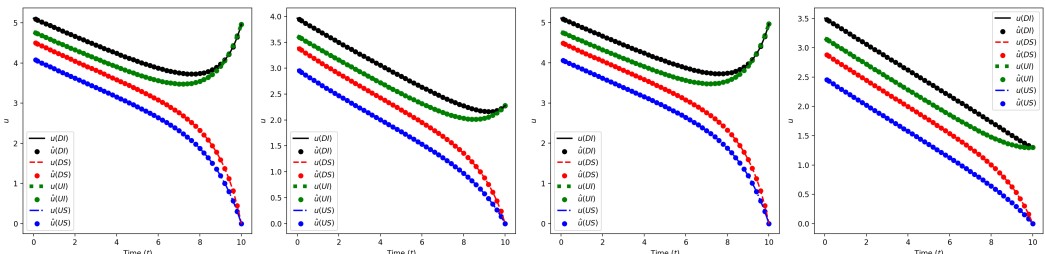

Figure 11: Learned value function for four randomly sampled pairs $(\eta, \kappa)$, with $\kappa \in [0, 10]$, along a time discretization with $M = 50$ points for the cybersecurity model. Points indicate the approximate solution and curves indicate the true solution obtained via Picard iteration.

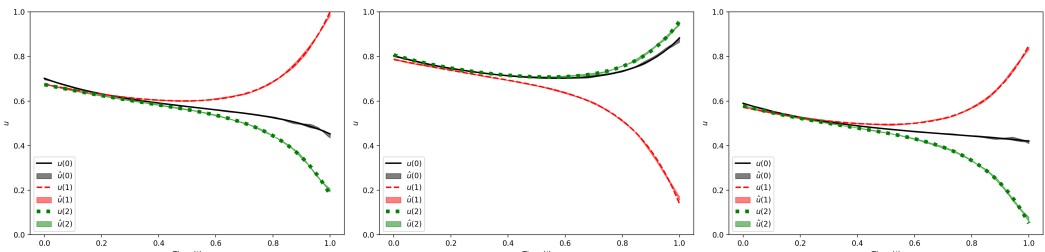

Figure 12: Learned value functions for three randomly sampled pairs $(\eta, \kappa)$, denoted by $\widehat{u}$, approximating the flow map $\Phi$ for a $d = 3$ dimensional quadratic model, for three random initial distributions and parameters $\kappa \in [0, 1]^3$ sampled uniformly at random. Averages are taken across 5 trials, and shaded regions on approximate curves present error bars of one standard deviation above/below the mean across trials.

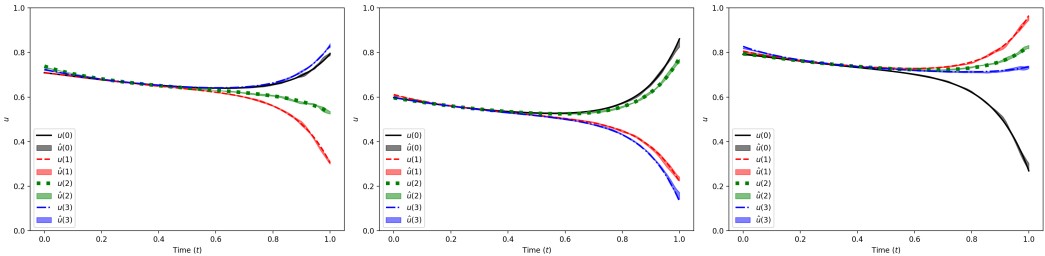

Figure 13: Learned value functions, denoted by $\widehat{u}$, approximating the flow map $\Phi$ for a $d = 4$ dimensional quadratic model, for three random initial distributions and parameters $\kappa \in [0, 1]^4$ sampled uniformly at random. Averages are taken across 5 trials, and shaded regions on approximate curves present error bars of one standard deviation above/below the mean across trials.

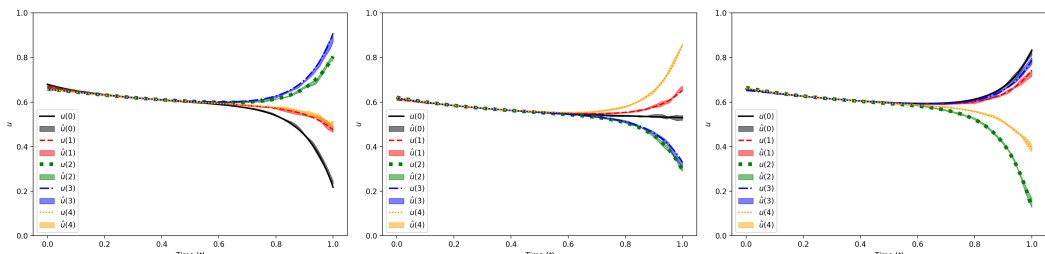

Figure 14: Learned value functions, denoted by $\widehat{u}$, approximating the flow map $\Phi$ for a $d = 5$ dimensional quadratic model, for three random initial distributions and parameters $\kappa \in [0,1]^5$ sampled uniformly at random. Averages are taken across 5 trials, and shaded regions on approximate curves present error bars of one standard deviation above/below the mean across trials.

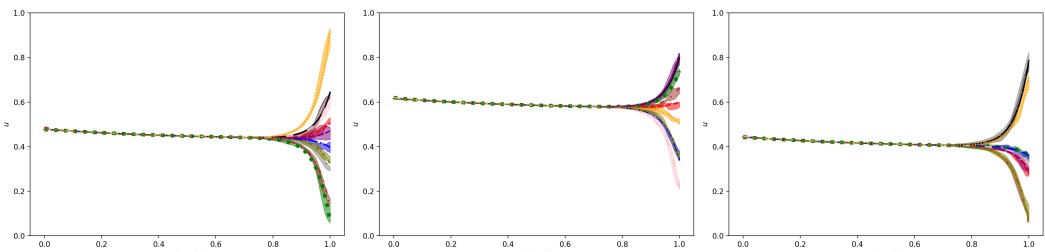

Figure 15: Learned value functions, denoted by $\widehat{u}$, approximating the flow map $\Phi$ for a $d = 10$ dimensional quadratic model, for three random initial distributions and parameters $\kappa \in [0,1]^{10}$ sampled uniformly at random. Averages are taken across 5 trials, and shaded regions on approximate curves present error bars of one standard deviation above/below the mean across trials.

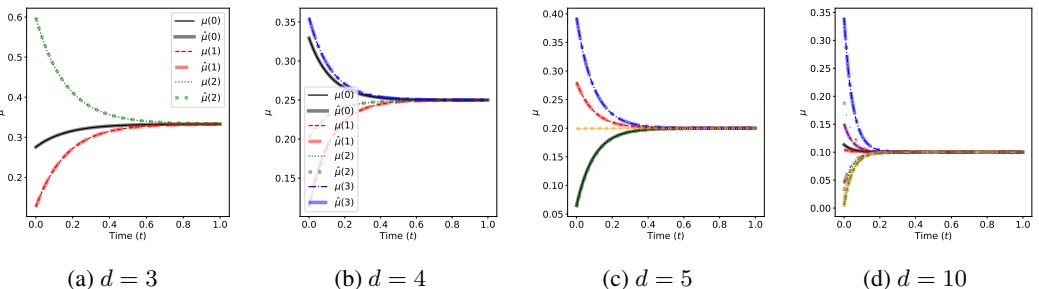

(a) $d = 3$        (b) $d = 4$        (c) $d = 5$        (d) $d = 10$

Figure 16: Comparison of true flows of measures $\mu$ and learned flows of measures $\widehat{\mu}$ for randomly sampled pairs $(\eta, \kappa)$ in dimensions $d = 3, 4, 5, 10$ respectively.

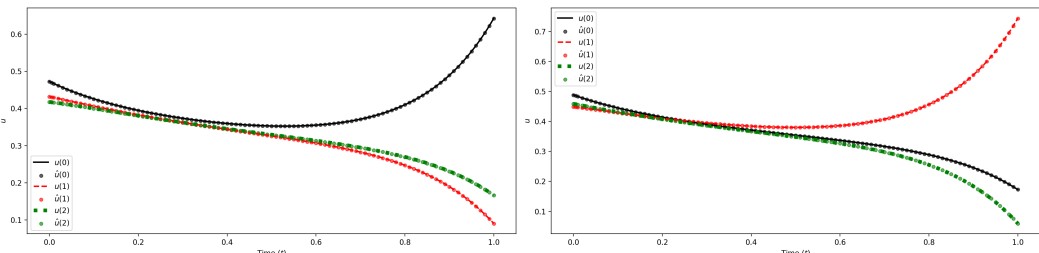

Figure 17: Learned value function for two randomly sampled pairs $(\eta, \kappa)$, along a time discretization with $M = 100$ points in dimensions $d = 3$. Points indicate the approximate solution and curves indicate the true solution obtained via Picard iteration.

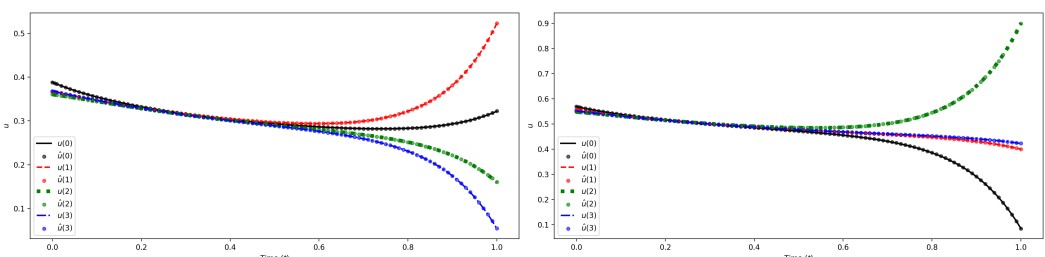

Figure 18: Learned value function for two randomly sampled pairs $(\eta, \kappa)$, along a time discretization with $M = 100$ points in dimensions $d = 4$. Points indicate the approximate solution and curves indicate the true solution obtained via Picard iteration.

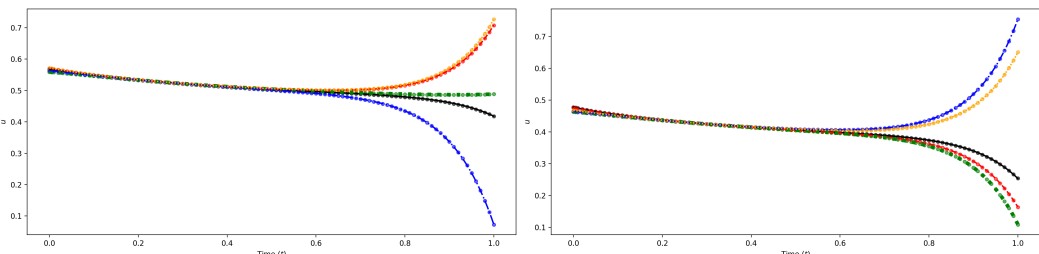

Figure 19: Learned value function for two randomly sampled pairs $(\eta, \kappa)$, along a time discretization with $M = 100$ points in dimensions $d = 5$. Points indicate the approximate solution and curves indicate the true solution obtained via Picard iteration.

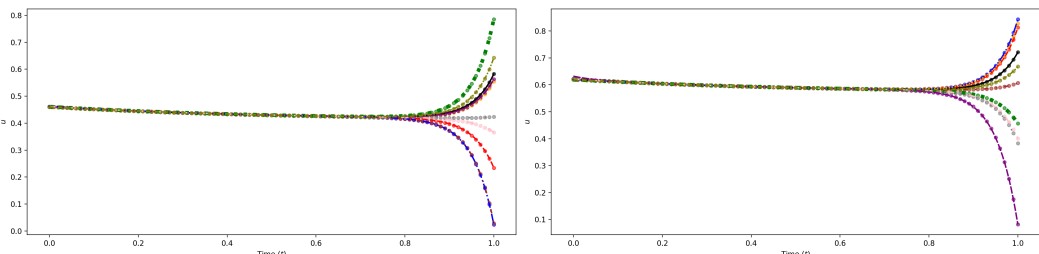

Figure 20: Learned value function for two randomly sampled pairs $(\eta, \kappa)$, along a time discretization with $M = 100$ points in dimensions $d = 10$. Points indicate the approximate solution and curves indicate the true solution obtained via Picard iteration.

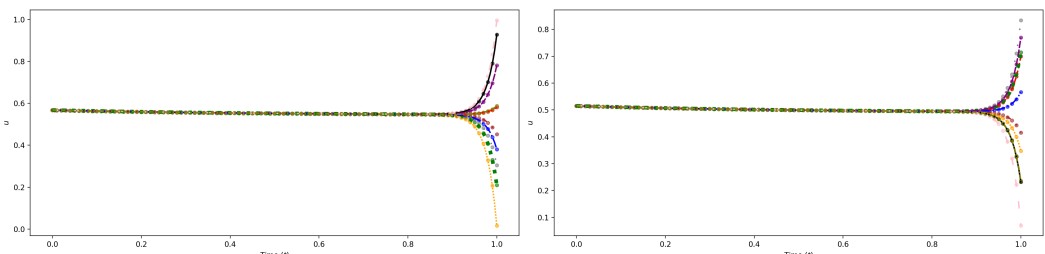

Figure 21: A slice of 10 components of the learned value functions, for two randomly sampled pairs $(\eta, \kappa)$, along a time discretization with $M = 100$ points in dimensions $d = 20$. Points indicate the approximate solution and curves indicate the true solution obtained via Picard iteration.

