# OpenReview forum: "Operator Learning for Families of Finite-State Mean-Field Games"
_ICLR.cc/2026/Conference — Submitted to ICLR 2026_

### Official Review · Reviewer_Di3U · 2025-10-21

**Soundness:** 3
**Presentation:** 3
**Contribution:** 3
**Rating:** 6
**Confidence:** 3

**Summary:**

This work proposes an operator learning method to approximate Nash equilibrium (NE) of families of finite-state mean-field games (MFGs), which are represented using forward Kolmogorov-Fokker-Planck (KFP) equation. Unlike standard solution techniques which require re-solving the game for any change in the configuration, learning mapping between the distribution of configurations to the corresponding solution (here, NE) is greatly efficient and desired. Motivated by this, the authors frame MFG equilibria as outputs of an operator, dubbed flow map, which maps the initial distributions (of initial states) and cost functions to its corresponding NE.

**Strengths:**

The paper is well written with necessary proofs to support the claims. Operator learning is a natural progression when it comes to problems of this nature (i.e., learning equilibrium control policies, solving PDEs), and there exists a relevant literature that proposes operator learning for games (although it is for differential games) [1]. This brings us to the weakness.

[1] Zhang, L. et al., *Pontryagin Neural Operator for Solving General-Sum Differential Games with Parametric State Constraints*. L4DC 2024.

**Weaknesses:**

I think the authors need to highlight the challenges in extending existing operator learning methods to solve MFGs; otherwise it risks reading as merely a new case study, which I don't believe is the case here.

**Questions:**

Some questions and comments:

1. It seems that the bottleneck is the picard solver, can the authors comment on the following queries:

    - How well does the current picard iteration implementation handle high dimensional problems (i.e., how does the empirical complexity look like)?
    - How do you obtain the initial guess and how sensitive is the current solver to the initial guess?
    - In your experiments, do you notice any convergence issues?

2. Do you have any insights on why ResNet results in better performance?

3. While value function plots are informative, it would be better if the authors could show additional visualization showcasing the interaction (e.g., state evolution). This might help readers familiarize with the examples and make sense of the optimal actions.

4. You mention that learning becomes unstable beyond d=10 in the case of high-dimensional quadratic models. Could you comment on which aspect of the algorithm leads to the instability? Is it the picard solver?

5. Just curious if the authors tested out-of-distribution generalization of the learned flow operator.

---

> ### Author Response · Authors · 2025-11-21
>
> We thank the reviewer for their time and feedback on our work. Below, we provide responses to the weakness and questions raised in your review, and we hope that we can clarify any confusion. We are also happy to provide additional discussion and/or experiments to resolve remaining doubts about our work.
>
> **Weakness: Challenges with Using Standard Operator Learning Approaches for Finite-State MFGs**
>
> One could, in principle, apply a black-box operator learning model such as DeepONet or FNO to the master equation, the nonlinear PDE that governs the MFG system (see Appendix C of our paper for more details on the master equation). However, as pointed out in [1], such operator learning frameworks have a serious theoretical pitfall: they are prone to a parametric curse of complexity, meaning that they may require exponentially many parameters (in the approximation error) to approximate the solution to a nonlinear PDE to a reasonable degree of accuracy.  We aim to highlight this issue, as well as the motivation for our distinct approach in Lines 87–100 of our paper. If we can make the necessity of our novel method more apparent in other sections as well, please let us know, and we will incorporate it into our revision.
>
> Although modern operator learning models are powerful, it can be quite difficult to extract theoretical guarantees such as our Corollaries 4.3 and 4.5 for complicated nonlinear PDE like the master equation. Overall, our method is the first foray into operator learning for finite-state MFGs, and our primary goal is to provide a straightforward, theoretically motivated, and robust approach for solving such MFGs. Along the way, we provide a novel regularity guarantee for flow maps of families of finite-state MFGs (Theorem 4.1) that allows us to establish the first generalization guarantee for learning finite-state MFGs, at least to our knowledge.
>
> **Questions 1: Complexity of Picard Iteration**
>
> Although solving higher-dimensional problems via Picard iteration certainly takes longer, our Picard solver remains remarkably robust for generating samples, even for higher-dimensional examples. By default, we initialize Picard iteration with $\mu(t, x) = \eta(x)$ and $u(t, x) = g_\kappa (x, \eta(x))$ for all $t$ along a given time discretization, where $(\eta, g_\kappa)$ is the pair of initial distribution and final cost for which we are trying to generate a sample.
>
> Even with an extremely low tolerance of $\varepsilon = 10^{-10}$ between two consecutive fixed point iterations when generating data for the $d = 20$ quadratic model, we find that Picard iteration generally converges after approximately five steps, in about a second, for generating a single sample. Of course, this process takes significantly longer when generating thousands of sample trajectories as training data for our neural operator, but generating each sample is quite efficient, even in higher dimensions; we have yet to observe any issues with convergence. When we change the initialization, by taking $\mu(0, x) = \eta(x)$ and $u(T, x) = g(x, \eta(x))$ and $\mu(t, x) = u(t, x) = 0$ for all other $t$, for example, Picard iteration still converges rapidly, only taking one to two more iterations on average to converge. If there is interest, we can include some of these experiments in an appendix of our revised paper to illustrate the efficiency of our sampling procedure.
>
> **Question 2: Insight Into ResNet Performance**
>
> We expect that the performance of ResNet comes from its ability to stabilize the training procedure via skip connections. Indeed, we observed that for higher-dimensional examples, the optimization trajectories for feedforward ReLU networks became increasingly unstable, pointing us towards a choice of architecture that, at least heuristically, can provide more stable training. This heuristic choice, is well-supported. For instance, [2] shows that ResNets and other architectures with skip connections can alleviate the issue of shattered gradients and have smoother loss landscapes when compared to feedforward architectures. As a result, convergence to the empirical risk minimizer (or, at least, a nearby set of parameters) is faster and more reliable with ResNet. Otherwise, based on our theoretical guarantees, moderately sized feedforward ReLU neural networks are sufficiently expressive to learn flow maps for finite-state MFGs, and we believe that our experiments largely support this conclusion.
>
> **References**
>
> [1] Lanthaler, S., and Stuart, A. (2025). "The parametric complexity of operator learning." IMA Journal of Numerical Analysis.
>
> [2] Balduzzi, D., Frean, M., Leary, L., Lewis, J. P. , Ma, K.W., and McWilliams, B. (2017). "The Shattered Gradients Problem: If resnets are the answer, then what is the question?" In Proceedings of the 34th International Conference on Machine Learning (pp. 342–350). PMLR.

---

> ### Author Response · Authors · 2025-11-21
>
> **Question 3: Plots of State Distributions**
>
> We thank the reviewer for pointing this out; we agree that it may help readers to see the flow of measures that corresponds to a given value function at the MFG equilibrium. In our revised paper, we have included visualizations of the flow of measures for both the cybersecurity example and the quadratic model. For the former, each state has a natural interpretation (e.g., defended and infected, susceptible and uninfected, and so on), so we have included the visualization in the main body of the paper. For the latter, the model is abstract and thus the flow of measures is less interpretable, especially in high dimensions. Due to this and space constraints, we have included the corresponding plot for the quadratic model for the in Appendix G.
>
> **Question 4: Training Instability in High Dimensions**
>
> As noted above, the Picard solver (i.e., the source of samples for our operator learning problem) is quite stable, even in higher dimensions. Rather, it is the neural network training procedure that becomes unstable. By switching to ResNet, an architecture with smoother loss landscapes, we are able to partially mitigate this issue in $d = 10$, and we expect that similar architectural changes may yield better results for other high-dimensional tests. However, we emphasize that, even in dimension $d = 10$, tools from the standard optimization toolbox (e.g., using weight decay and momentum, changing the loss function to smooth $L^1$ loss, and so on) are sufficient to produce accurate results using only feedforward ReLU networks.
>
> **Question 5: Out-of-distribution Generalization**
>
> During our work on this project, we did consider the out-of-distribution generalization of our method. The specific setting that we used was the cybersecurity model, trained on samples with terminal cost of the form $g_\kappa(x, \eta) = \kappa \mathbf{1}_{\{DI, UI\}}(x)$ for $\kappa \in [0, 1]$ and tested on samples with terminal cost parameter $\kappa \in [0, 10]$. We did not observe very impressive performance in this setting, and as a result, did not pursue the question of out-of-distribution generalization further. From our theoretical results alone, it is difficult to justify why our method (or any operator learning method that reduces the problem of learning flow maps to regression, for that matter), would be expected to perform well out-of-distribution. However, we agree that it is an important question, especially in cases where it is only easy to generate samples for costs belonging to a subset of family of terminal costs. As a result, we view out-of-distribution generalization as an interesting area of future work, building upon the foundation we provide in this paper.

---

> > ### Comment · Reviewer_Di3U · 2025-11-27
> > **Acknowledgement**
> >
> > Thank you for addressing my concerns and highlighting the theoretical guarantees associated with the proposed method, which may not be easily established with the existing Operator Learning methods. The authors have also added the state distribution plots, which will be of great help to the readers. In light of these, I have raised my score to a 8.

---

> > > ### Author Response · Authors · 2025-11-27
> > >
> > > We thank the reviewer again for their helpful comments, feedback, and the score update! We are glad that we have clarified some of the contributions of our work and that the additional visualizations of the flow of measures make our experiments more interpretable. If there are any remaining concerns that we can address during the remainder of the discussion period, please let us know.

---

### Official Review · Reviewer_PKPa · 2025-10-30

**Soundness:** 3
**Presentation:** 2
**Contribution:** 2
**Rating:** 4
**Confidence:** 3

**Summary:**

This paper studied the proposed operator learning method of numerically solving finite-state mean-field games (MFGs). The main contributions include:

* A framework of solving MFGs by using neural networks to approximate the flow map, which is an operator that maps the initial distribution of players’ states $\eta$ and the terminal cost function $g$ to the value function $u$. The framework generally follows the operator learning methods in the literature. Compared with previous methods, the advantage of the proposed method is that it can treat all the initial conditions without re-training.

* The authors provide theoretical analysis on the generalization error and approximation error of the proposed method. Especially, the analysis provides guarantees on the errors in terms of the weights of the networks, the state space dimension, and the dimension of the space that is used to parameterize the terminal cost function.

* Two numerical experiments are provided to show the efficiency of their methods: the cybersecurity model with the dimension of state space $4$, and quadratic models with 10 dimensions. In these experiments, the authors compare the output of their operator learning methods with the solutions produced by solving the ODE systems numerically.

**Strengths:**

* The studied problem is well-motivated, and the relationship between the current work and related works is clearly stated.
* The experimental settings are described in detail.
* The authors have provided both theoretical analysis and experimental results on their proposed methods.
* The proposed method has the potential to be used to solve MFGs efficiently, because it does not need to be re-trained to adapt to a specific choice of the MFG.

**Weaknesses:**

**Theoretical side**: I think the theoretical bounds provided in this paper are loose, and I doubt whether it is possible to use them to gain practically relevant insights. Especially, my concerns mainly rely on the following points:

* Both in Corollary 4.3 and 4.5, the dependency on the parameter $\mathcal{K}$ is omitted in the big-O notation. Since the dimension of the parameterized space of terminal cost functions is an important block of the proposed methods, and its dimension can be large, I think it is important to have a clear understanding of how its dimension affects the performance of the proposed method.

* As the authors said in Appendix E.2, Corollary 4.3 and 4.5 are almost directly taken from the corresponding results in (Jiao et al., 2023). However, the results of (Jiao et al., 2023) are very general results that hold for ReLU neural networks on regression problems. This means the bounds in Corollary 4.3 and 4.5 in the current paper do not use the special structures of MFGs. This makes me feel the provided bounds are too general and cannot provide special insights into the problem of solving MFGs with neural networks.

**Experimental side**: My general feeling is that the scope of experiments in this paper is rather limited: only two examples are provided, and how the choices of architectures, optimizers, sampling method, and loss functions affect the performance is discussed very briefly.

For example, most experiments are done with fully connected ReLU networks, and it is only briefly mentioned that the ResNet architecture can benefit the performance when $d = 10$ in the quadratic models. It is hard to gain understanding from such a brief discussion, and it may lead to confusion: does the ResNet architecture only have better performance than fully connected ReLU networks when $d = 10$, or does such a phenomenon consistently appear in other settings?

The authors motivate their choice of using fully connected ReLU neural networks as being most aligned with their theoretical results (lines 680–682). However, since these experiments are not only aimed at verifying their theoretical results, I am not convinced that this is a good reason to limit their experiments to fully connected ReLU networks. Even for the purpose of verifying their theoretical bounds, I also did not see any discussion on the comparison between their theoretical results and experimental results.

**Other problems**:
* In the first paragraph of Section 2.1, there is no clear definition of what the "action" of the players is.
* In line 153, the notation $\mathcal{L}(\cdot)$ is not defined.
* In Figures 2 and 3, the trajectories of the proposed methods and the ODE solvers are too close. I suggest including figures that present their differences as well.

**Questions:**

* Compare with the general bounds provided in (Jiao et al., 2023), what special insights can Corollary 4.3 and 4.5 provide for the problem of learning flow maps using neural networks?
* I suggest that the authors add discussions on how the architectures, optimizers, sampling methods, and loss functions can affect the performance of the proposed methods.

---

> ### Author Response · Authors · 2025-11-21
>
> We thank the reviewer for their time and feedback on our work. Below, we provide responses to all of the weaknesses and questions in your review, and we hope that we clarify any confusion. We are also happy to discuss any points in greater detail and/or provide additional numerical to address remaining concerns.
>
> **Weakness 1: Theoretical Side**
>
> **Dependence of Theoretical Results on Dimension of Parameter Set**:
> We would like to clarify that the dependence on the space of terminal costs Corollary 4.3 and Corollary 4.5 is present. In both results, $\mathcal{K}$ denotes the space of parametrized terminal costs, which we assume has dimension $k$. In contrast, $K$ instead refers to the depth of the neural network that we are using to approximate the flow map, in line with the notation from [1] We make the dependence on $k$, the dimension of the space of terminal costs, explicit in both of our theoretical results.
>
> **Leveraging Structure of MFGs**:
> It is true that the results of [1] apply to regression with ReLU networks. However, in order to reach a point where we can apply approximation and generalization results in the first place, we heavily leverage MFG structure to show that the flow map induced by a parametrized family of finite-state MFGs is sufficiently regular. This novel regularity result is presented in Theorem 4.1, and the corresponding proof can be found in Lemma E.1 through Lemma E.4. More fundamentally, our method is motivated by the philosophy of operator learning: reduce to a simpler regression problem, with samples given input-output pairs from the operator in question. To our knowledge, we provide the first work exploiting the structure of the forward-backward MFG system (rather than the master equation) to solve finite-state MFGs, and couples with our theoretical analysis of the regularity of the flow map, the results of [1] allow us to extract concrete approximation and generalization guarantees for this setting.
>
> **References**
>
> [1] Jiao, Y., Wang, Y., and Yang, Y. "Approximation bounds for norm constrained neural networks with applications to regression and GANs." Applied and Computational Harmonic Analysis, 65:249–278, July 2023.

---

> ### Author Response · Authors · 2025-11-21
>
> **Weakness 2: Experimental Side**
>
> **Scope of Experiments**:
> For an instance of another recent paper that considers the problem of learning finite-state MFGs, please see [2]. We present numerical experiments on the same set of baselines: the $d = 4$ dimensional cybersecurity model and the quadratic cost model. However, our experiments in the quadratic cost case extend to dimension $d = 10$, whereas most other works are limited to $d = 3$. We provide more extensive discussion of our experimental procedure in Appendix A, including the selection of architecture, optimizer, and loss function is discussed therein. However, we agree that our initial discussion in Section 5 of our paper was a bit cursory. As a result, we have expanded upon our experimental procedure significantly in the main body of the revised paper, providing more intuition for why ResNet may perform better in dimension $d = 10$ and describing our selection of optimizer, loss, and architecture in more detail. Our sampling procedure is intrinsic to Algorithm 1, so there is not much that we can test/change in that regard.
>
> **Examples**:
> If the reviewer is curious, we are likely able to implement an additional example demonstrating the accuracy of our method. For instance, [3] provides a class of two-state MFGs for socio-economic models that may act as another reasonable test case for finite-state MFG solvers. If this would be of interest, please let us know, and we can upload additional experiments for this class of MFGs.
>
> **Architecture**:
> On the point of ResNet, we only found that adding skip connections (e.g., using a ResNet-like architecture) was beneficial in the case $d = 10$; this phenomenon is not apparent in lower dimensions. We posit that this is because the optimization landscape is smoother for networks with skip connections, as justified in [4]. In lower dimensions, the optimization procedure for fully-connected feedforward ReLU networks is sufficiently stable to the point that the benefit of adding skip connections is negligible. See the revised Section 5 of our paper for an updated commentary on this point.
>
> **Link with Theory**:
> Finally, we do intend for our numerical experiments to provide confirmation of our theoretical results, as seen in Figure 4 of our paper. There, we show that as the width of the network increases, we obtain better approximation error (as measured by training loss) and generalization performance (as measured by test loss). These observations are in agreement with Corollary 4.3 and 4.5, both of which predict that as we allow the weights and widths of our networks to grow while holding the depth constant, we should observe better approximation and generalization performance.
>
> **Weakness 3: Notation and Miscellaneous**
>
> **Definition of Action**: The "action" of the player refers to the chosen control, from the admissible set of controls outlined at the start of Section 2.1. We have cleaned up our language so that we only refer to controls throughout, rather than referring to controls and actions interchangeably.
>
> **Definition of $\mathcal{L}$**: In Line 153, the notation $\mathcal{L}$ refers to the law of a random variable. We have replaced it with $\mathrm{Law}(\cdot)$ to provide more clarity for readers.
>
> **Plots of Errors**: We agree that it may be informative for readers to see plots of the difference between the proposed methods and the ODE solvers. We have included a plot of this form in the body of our paper to enable readers to more accurately evaluate the performance of this method – see Figure 4 in the updated version of the paper, as well as Appendix G for additional plots displaying the error on the cybersecurity model.
>
> **References**
>
> [2] Cohen, A., Lauriere, M., and Zell, E. (2024). "Deep Backward and Galerkin Methods for the Finite State Master Equation." Journal of Machine Learning Research, 25(401), 1–50.
>
> [3] Gomes, D., Velho, R., and Wolfram, M.T. (2014). "Socio-economic applications of finite state mean field games." Philosophical Transactions of the Royal Society A: Mathematical, Physical and Engineering Sciences, 372(2028).
>
> [4] Balduzzi, D., Frean, M., Leary, L., Lewis, J. P. , Ma, K.W., and McWilliams, B. (2017). "The Shattered Gradients Problem: If resnets are the answer, then what is the question?" In Proceedings of the 34th International Conference on Machine Learning (pp. 342–350). PMLR.

---

> ### Author Response · Authors · 2025-11-21
>
> **Question 1: Insights from Theoretical Guarantees**
>
> While it is true that our corollaries are an application of the results from Jiao et al., we believe that they do still provide valuable insight into learning flow maps for entire families of finite-state MFGs. One insight that they provide is the fact that relatively shallow neural networks can learn flow maps to high accuracy, while the weight and the width of the networks must scale polynomially with the desired approximation error. Also, at least to our knowledge, we are able to provide the first sample complexity guarantee for learning finite-state MFGs in Corollary 4.5. Using a different approximation guarantee (e.g., from [5]) would not allow for such a sample complexity guarantee. To apply the results from Jiao et al. in the first place, we need to establish a regularity guarantee for flow maps of *families* of finite-state MFGs, presented in Theorem 4.1. The proof of this result is nontrivial, and may be of independent interest to those studying ML methods for MFGs.
>
> **Question 2: Additional Discussion of Experimental Choices**
>
> As noted above, in our revised paper, we now include some additional discussion of our choice of architecture, optimizer, and loss function in Section 5 of our revised paper.
>
> **References**
>
> [5] Yarotsky, D. (2017). "Error bounds for approximations with deep ReLU networks." Neural Networks, 94, 103–114.

---

> ### Comment · Reviewer_PKPa · 2025-11-27
>
> I thank the authors for their extensive explanation. Now I have a better understanding of their contribution, especially their efforts in proving the regularity guarantee for flow maps of families of finite-state MFGs. I have updated my rating to 6.

---

> > ### Author Response · Authors · 2025-11-27
> >
> > We thank the reviewer again for their helpful feedback, comments, and score update, and we are glad that our contributions have been clarified! If there are any other areas of concern that we can address prior to the end of the discussion period, please let us know.

---

### Official Review · Reviewer_v1jZ · 2025-11-01

**Soundness:** 3
**Presentation:** 3
**Contribution:** 3
**Rating:** 8
**Confidence:** 3

**Summary:**

This paper proposes an operator learning framework to solve entire families of finite-state mean-field games (MFGs). Instead of solving each MFG instance individually, the authors aim to learn a neural network that approximates the "flow map" as an operator that maps the game's parameters (initial player distribution and a terminal cost parameter) to the corresponding value function. The methodology involves generating training data by solving specific MFG instances using fixed point iteration on the coupled forward-backward HJB-KFP system. The paper provides theoretical guarantees on the approximation and generalization error of this approach, and the framework is is empirically validated on a cybersecurity model and a high-dimensional quadratic MFG benchmark.

**Strengths:**

- The primary strength of this work is its framing of the problem. Moving from solving single MFG instances to learning an operator for entire parameterized families is a significant and practical step forward. This approach enables rapid, retrain-free generalization to new initial conditions and cost functions, which is highly valuable for applicability of MFGs.

- The paper provides rigorous theoretical guarantees for the approximation error and generalization error. The results require common MFG assumptions.

- The paper is well-written, structured, and easy to follow. It begins with a clear introduction to finite-state MFGs, logically motivates the need for operator learning, defines the flow map, details the algorithm, and then presents the theory and experiments. The connection to prior work is clearly articulated.

- The experiments are well-chosen and effectively demonstrate the method's capabilities. The use of a standard cybersecurity benchmark and a scalable quadratic model shows the method's accuracy and performance as the state-space and parameter dimensions increase. The inclusion of Figure 4, which analyzes the effect of network width on training and test loss, is a good ablation study that supports the theoretical claims.

**Weaknesses:**

- The paper honestly notes that learning becomes "increasingly unstable" for dimensions beyond d=10. The success in d=20 is shown for a simplified setting with a fixed time discretization. This suggests that the primary method has practical scalability limits, since only 10 to 20 states could be too little in practice.

-  The theoretical guarantees, while valuable, also exhibit a "curse of dimensionality" in their dependence on the state. These bounds suggest that the methodology in general does not scale well.

**Questions:**

1.  The paper mentions instability for d > 10. Is this instability primarily a result of the data generation step (i.e., convergence issues with Picard iteration in high dimensions) or the neural network training process (e.g., optimization challenges)?

2.  Does the methodology generalize to discrete-time problems beyond time discretizations of continuous-time problems, or do you expect additional difficulties?

---

> ### Author Response · Authors · 2025-11-21
>
> We thank the reviewer for the time and feedback on our work. Below, we provide responses to the weaknesses and questions in your review, and we hope that we clarify any confusion. We are happy to discuss anything in greater detail and/or provide additional experiments to address remaining concerns as well.
>
> **Weakness 1: Scalability of Method**
>
> We agree that scalability of our operator learning method is an important aspect, and our work is intended to provide a firm theoretical foundation for operator learning approaches to finite-state MFGs. However, we believe that future empirical work (e.g., using more expressive architectures, as in the ResNet experiments in Appendix G of our paper) may yield better scaling to higher dimensions. Nonetheless, our experiments in dimensions $d = 10$ and $d = 20$ exceed the maximal dimension that prior ML-based approaches to MFGs by a factor of at least two (in Cohen et. al (2024), tests for the quadratic model are provided up to dimension $d = 3$, for example), all while extending to the setting of variable terminal cost. We also note that many works on neural operators do not consider high-dimensional examples; see for instance DeepONets [1] and Fourier Neural Operators [2], whose examples are two-dimensional. The main challenge is to learn the operator, even if the state-space is low-dimensional.
>
> **Weakness 2: Dimension-Dependence of Theoretical Guarantees**
>
> We agree with the reviewer that there is ample opportunity for future work regarding methods whose sample complexity scales better with the dimensionality of the underlying state space. However, our work presents the first characterization of sample complexity for an ML-based method for finite-state MFGs, and the dependence of the approximation error and sample complexity on dimension in Corollaries 4.3 and 4.5 is the same is in [3], where such scaling is presented as parameter-efficient operator learning  (although, as pointed out in Remark E.1, we use a different approach to obtain our theoretical results). As mentioned above, we also provide compelling numerical experiments for test cases in higher dimensions than in any related work in the finite state space setting, at least to the best of our knowledge.
>
> **Question 1: Stability of Picard Iteration**
>
> Instability in high dimensions is largely the result of optimization becoming increasingly difficult in high dimensions. Indeed, the data generation step, via Picard iteration, remains remarkably consistent across dimensions.
>
> **Question 2: Extension to Discrete-Time Problems**
>
> At a high level, we expect that our methodology does generalize to discrete time problems. We require two things to hold in order to apply our methodology:
>
> 1. For specific instances of the problem in question, data generation is efficient. In our setting, this means that it is relatively easy to generate samples for fixed initial distributions and terminal costs via Picard iteration. In a discrete-time setting, this may correspond to solving a large linear (or nonlinear) system.
> 2. The underlying dynamics of the problem possess enough regularity to be well-approximated by a moderately sized neural network. We prove this in Appendix E in our paper (our flow map is Lipschitz). In the discrete time setting, the analogue may look quite different, although we are aware of similar results for approximating sequences and discrete-time dynamical systems with recurrent neural networks (RNNs) for instance, as presented in [4].
>
> Under these conditions, we anticipate that future work may be able to extend our method quite naturally to other families of dynamical systems, including in the discrete time case.
>
> **References**
>
> [1] Lu, L., Jin, P., Pang, G., Zhang, Z., and Karniadakis, G. E. (2021). "Learning nonlinear operators via DeepONet based on the universal approximation theorem of operators." Nature machine intelligence, 3(3), 218-229.
>
> [2] Kovachki, N., Li, Z., Liu, B., Azizzadenesheli, K., Bhattacharya, K., Stuart, A., and Anandkumar, A. (2023). "Neural operator: Learning maps between function spaces with applications to PDEs." Journal of Machine Learning Research, 24(89), 1-97.
>
> [3] Lanthaler, S., and Stuart, A. (2025). "The parametric complexity of operator learning." IMA Journal of Numerical Analysis.
>
> [4] Gonon, L., Grigoryeva, L., and Ortega, J.P. (2023). "Approximation bounds for random neural networks and reservoir systems." The Annals of Applied Probability, 33(1).

---

> > ### Comment · Reviewer_v1jZ · 2025-11-26
> >
> > I thank the authors for their extensive response and update. My concerns regarding the limitations have been discussed and addressed by the response. Regarding exceeding prior results with $d=10$ and $d=20$, although I agree with the authors, I think again using the time discretization discussed by authors and something like Online Mirror Descent may work, which scaled up to trillions of finite states. I will keep my score.

---

> ### Author Response · Authors · 2025-11-27
>
> We thank the reviewer again for their useful comments and feedback!
>
> We agree that exploring iterative methods such as online mirror descent (OMD), coupled with the version of our method that learns the flow map along a fixed time discretization, may be a promising avenue for future research. However, we still believe that there is a significant gap between previous research using OMD for solving finite-state MFGs and our approach. For instance, although [1] shows that OMD is a viable strategy for solving high-dimensional finite-state MFGs, their work is limited to the setting of fixed initial distributions, terminal costs, and focuses on discrete-time MFGs (rather than continuous-time MFGs, described by the dynamics of the master equation, as in our work). Extending OMD into an operator learning method that can reliably predict value functions for entire families of MFGs, with variable initial distributions and terminal costs, would be a very interesting, albeit potentially challenging, line of research. As it stands, we believe that the method that we propose in our paper is the most natural approach for simultaneously computing equilibria of families of continuous-time, finite-state MFGs.
>
> [1] Pérolat, J., Perrin, S., Elie, R., Laurière, M., Piliouras, G., Geist, M., Tuyls, K., and Pietquin, O. (2022). "Scaling Mean Field Games by Online Mirror Descent." In *Proceedings of the 21st International Conference on Autonomous Agents and Multiagent Systems* (pp. 1028–1037). International Foundation for Autonomous Agents and Multiagent Systems.

---

### Official Review · Reviewer_rSss · 2025-11-01

**Soundness:** 2
**Presentation:** 3
**Contribution:** 2
**Rating:** 4
**Confidence:** 3

**Summary:**

This paper studies a supervised operator learning method for solving mean-field games (MFGs) with finite state spaces. The proposed method trains a neural operator using samples generated by Picard iteration. It then applies the neural operator to solve different MFGs with varying initial distributions and terminal costs. Theoretical guarantees are discussed, and numerical experiments are provided.

**Strengths:**

- This work is well-motivated, as learning MFGs is in general challenging and computation-heavy.
- The paper is well-written and easy to follow.
- Theoretical guarantees are discussed for the supervised learning method.
- Various numerical experiments are conducted, with their settings and results clearly presented.

**Weaknesses:**

- Any finite state space can be embedded into a continuous state space. It's thus not clear to me why addressing only finite state spaces is a contribution of this work over those addressing continuous state spaces.
- The contribution of the work over Cohen et al. (2024) is the ability of handling different terminal cost functions, which seems marginal to me.
- Separable running cost function (action and population are decoupled) is considered, which is a huge simplification that needs further clarification.
- The excess risk misses two significant error sources: discretization error and sample label approximation error. The latter is critical: for each "sample", its reference equilibrium is only the approximate solution generated by finite steps of Picard iteration.
- The approximation error of the neural network depends on $K$, the bound of the weights. How large is $K$? How do we know if it's bounded? To give an informative error bound, $K$ needs to be expressed using algorithm parameters.
- Typos and format issues.
  - Please correctly use in-text and end-of-text citations.
  - Line 056: typo "parame"
  - Line 061: has "a" generalization error
  - Line 260: $u^\eta$ is missing a superscript $g$.
  - Lines 266-267: $b_j$ and $B_j$ are inconsistent.
  - Line 347: the domain and codomain of $\Phi$ is off.

**Questions:**

See Weaknesses.

---

> ### Author Response · Authors · 2025-11-21
>
> We thank the reviewer for their time and feedback on our work. Below, we respond to some of the weaknesses mentioned in your review. We hope that our responses clarify any confusion, and we are happy to provide additional discussion and/or experiments to address remaining doubts about our work.
>
> **Weakness 1: Finite vs. Continuous State Spaces**
>
> While it is true that the state space of finite state mean-field game can be embedded into a continuous space, the dynamics of a finite-state MFG are very different than those of an MFG with a continuous state space. For instance, the master equation of a finite state space MFG is a PDE on the simplex, as outlined in Appendix C of our paper, while the representative agent's dynamics follow a Markov chain over a discrete space, as in Appendix B. However, the master equation in the continuous state space setting is a PDE on an appropriate (and infinite dimensional) space of probability measures, and the representative agent follows a diffusion process. It is possible to view the continuous setting as an appropriate limit of a sequence of finite-state MFGs, but this viewpoint is largely theoretical and has yet to provide practical algorithms for computing MFG equilibria. When the finite state space of an MFG has small cardinality (as is the case in many models of interest, including our numerical experiments), the limiting continuous state space model is essentially incomparable; see [1] for details on this notion of a limit and why the discrete and continuous state space settings are considered to be distinct models in the MFG literature. In summary, the underlying structure and dynamics of the two settings are fundamentally different, and approaches to solve MFGs in the two cases must reflect this difference.
>
> **Weakness 2: Contribution Compared to Cohen et al. (2024)**
>
> The authors of Cohen et al. (2024) provide a comprehensive analysis of two methods for solving single instances of an MFG via the master equation. However, neither method can learn dependence of a family of MFGs on the terminal cost or other model parameters, and as a result, our method is a significant generalization, solving entire families of MFGs with a *single* neural network.
>
> We also believe that our work provides a robust foundation for applications of operator learning to finite-state MFGs, with significant theoretical and practical benefits. While we instantiate the connection between operator learning and ML-based methods for finite-state MFGs by considering families of MFGs with varying terminal costs (and initial distributions), the framework that we provide is much more general. For instance, as we note in our conclusion, extending our method to families of MFGs with variable running costs is within reach, only requiring a regularity guarantee akin to our Theorem 4.1 to justify rigorously. Beyond this, in Corollary 4.3 and 4.5, we provide the first explicit parametric and sample complexity guarantees for any ML-based method for finite-state MFGs, operator learning or otherwise.
>
> **Weakness 3: Separable Running Cost**
>
> While this assumption may seem restrictive, it is necessary in order to obtain well-posed MFG systems with unique solutions. The argument behind this assumption is presented in Section 7.2.2 of *Probabilistic Theory of Mean Field Games with Applications: I* by Carmona and Delarue [2], and it remains standard for most analyses of finite-state MFGs. For instance, in Assumption 3.1, Section 3 of [3], which presents a thorough analysis of finite-state MFGs via the master equation, the same assumption is made. The separability assumption is also presented in Section 2.4 of [4], another important recent work in the finite-state MFG literature. MFG uniqueness plays an important role in Theorem 4.1, our regularity result for flow maps of families of finite-state MFGs, but we do not invoke this property elsewhere in our work.
>
> **References**
>
> [1] Bertucci, C., and Cecchin, A. (2024). "Mean Field Games Master Equations: From Discrete to Continuous State Space." SIAM Journal on Mathematical Analysis, 56(2), 2569–2610.
>
> [2] Carmona, R., and Delarue, F. (2018). *Probabilistic Theory of Mean Field Games with Applications I*. Springer International Publishing.
>
> [3] Bayraktar, E., and Cohen, A. (2018). "Analysis of a Finite State Many Player Game Using Its Master Equation." SIAM Journal on Control and Optimization, 56(5), 3538–3568.
>
> [4] Cecchin, A., and Pelino, G. (2019). "Convergence, fluctuations and large deviations for finite state mean field games via the Master Equation." Stochastic Processes and their Applications, 129(11), 4510–4555.

---

> ### Author Response · Authors · 2025-11-21
>
> **Weakness 4: Discretization Error and Excess Risk**
>
> The excess risk in Line 363 (now Line 367 in the revised paper) and the associated theoretical results in our paper are presented assuming that we have access to input-output pairs from the flow map, as is standard in operator learning. When describing our algorithm and in our numerical experiments, we use Picard iteration to generate these samples because it is a simple, robust, and efficient method. With a sufficiently small time discretization and a small enough tolerance for the difference between fixed point steps in Picard iteration (we take this to be $\varepsilon = 10^{-6}$ in our numerical experiments), we obtain very accurate approximations of the true flow map (e.g., with the error due to discretization much smaller than the optimization error mentioned in our theoretical guarantees). In principle, one could replace Picard iteration with any solver, and our theoretical results would still hold. In Section 4 of [5], the paper that proposed HJ-Nets as a parameter-efficient operator learning method for solving HJB equations and the original motivation for our method, the same assumption regarding the samples is made.
>
> **Weakness 5: Approximation Error and Weight Bounds**
>
> Corollary 4.3, which provides the approximation error of our method, states that for any $K \geq 1$ and $\varepsilon > 0$, there exists a network $\Phi$ with width $W = O(K)$, depth $L = O(\log(d + k))$, and weight bound $K$ such that
> \begin{align*}
> \|\Phi - \Psi\|_{\mathcal{C}([0, T] \times \mathcal{P}([d]) \times \mathcal{K})} \leq O(K^{-1/(d + k + 2)}) + \varepsilon.
> \end{align*}
> In other words, in order to achieve error on the order of $\varepsilon$, where $\varepsilon > 0$ is arbitrary, we can choose $K = O(\varepsilon^{-(k + d + 2)})$. This precisely quantifies the necessary weight bound, in terms of only $d$ and $k$, for our approximation guarantee to hold. From this, we can see that the requisite weight bound and width scales with the desired approximation error $\varepsilon > 0$, but the depth $L$ is independent of $\varepsilon$. This estimate is likely the best that we can hope for when using ReLU neural networks to approximate a Lipschitz map. Indeed, Theorem 5.1 in [5] presents an analogous scaling limit of $W = \tilde{O}(\varepsilon^{-(2d + 1)})$ on the width of HJ-Nets required to solve HJB equations to accuracy $\varepsilon > 0$, following from the ReLU network approximation result in [6].
>
> **Weakness 6: Typos and Formatting**
>
> We are very grateful to the reviewer for their close reading of our paper; we have corrected all of the above typos and formatting issues. For the last point, about the domain and codomain of $\Phi : [0,1]^d \to \mathbb{R}$ in Line 347 (now Line 350), we present the statement of Propositions 4.2 and 4.4 as they appear in Jiao et al., for regression with scalar labels, with inputs in the unit cube $[0, 1]^d$. In our Corollaries 4.3 and 4.5, we extend these results to the setting of vector-valued regression (which results in an extra factor of $d$ in the width of the networks) and apply an appropriate rescaling argument to pass from the unit cube to the domain of the flow map that we consider. These bookkeeping arguments are now presented in greater detail in Appendix E, and we have made this subtlety explicit in Section 4 of our paper as well.
>
> **References**
>
> [5] Lanthaler, S., and Stuart, A. (2025). "The parametric complexity of operator learning." IMA Journal of Numerical Analysis.
>
> [6] Yarotsky, D. (2017). "Error bounds for approximations with deep ReLU networks." Neural Networks, 94, 103–114.

---

### Author Response · Authors · 2025-11-21
**Revised PDF Uploaded**

First, we would like to thank all of our reviewers for the time that they spent reviewing our paper and the helpful feedback that they provided on our work during their initial reviews.

Second, we would like to announce that we have uploaded a **revised version** of our paper, taking into account the comments that we received from all of our reviewers. **Importantly, we have marked all revisions to the paper in red to indicate our changes.** Although the vast majority of the paper remained the same, two notable changes include:

1. **Additional Experiments and Visualizations**:
In Section 5 and Appendix G we now include plots of the flow of measures for both examples that we test our method on, showing that our method can also be used to recover very accurate approximations of flows of measures at MFG equilibria, in addition to the value function at the MFG equilbrium. We also provide plots of the error between the approximate and true value function for both examples, illustrating the accuracy of our method more explicitly than before.

2. **Discussion of Architectural and Optimization Choices**:
We include a more in-depth discussion of how we chose architectures, optimizers, and loss functions for our numerical experiments in Section 5, in addition to the full description of our experiments in Appendix A. Additionally, we expand upon our previous assertion that using an architecture with skip connections (e.g., ResNet) may yield slightly better performance during training.

We have also attempted to address all reviewer comments/questions on other parts of the paper in our revision, resulting in a handful of more minor changes throughout.

---

### Author Response · Authors · 2025-12-03
**Summary of Discussion with Reviewers**

As the discussion period draws to a close, we would like to summarize the outcomes of our productive conversations with each of our four reviewers. Additionally, we thank all of our reviewers again for their time and valuable feedback on our work!

**Reviewer rSss:** We addressed several questions pertaining to the assumptions of our work, including the distinction between MFGs with finite and continuous state spaces, the contribution of our work compared to previous methods for solving finite-state MFGs, and the importance of a separable running cost in order to ensure the uniqueness of MFG equilibria. We also clarified the significance of our theoretical results, justifying the exclusion of discretization error in our theoretical analysis of our method and explicitly quantifying the dependence of approximation guarantee on a bound $K$ on the weights of the approximating neural network. Reviewer rSss was unable to participate in the discussion before reviewer comments were disabled, but we believe that we were able to address all of their comments in detail in our initial response.

**Reviewer v1jZ:** We discussed the scalability of our method, demonstrating that our method provides accurate numerical results for state spaces whose dimensionality greatly exceeds that of previous approaches for solving finite-state MFGs with variable initial distribution (typically via the master equation). Next, we explained that the dimension dependence of our theoretical guarantees is in line with the best-known results on approximation and sample complexity bounds for approximating arbitrary Lipschitz functions with ReLU networks. At the urging of Reviewer v1jZ, we also explored how our method can extend naturally to discrete-time problems in our response. **Reviewer v1jZ was satisfied with our response and maintained their score of 8.**

**Reviewer PKPa:** We addressed Reviewer PKPa's concerns on both the theoretical and experimental side. On the theoretical side, we explained in greater detail why our theoretical results are novel, requiring a regularity result for flow maps of parametrized families of finite-state MFGs (Theorem 4.1 and Lemmas E.1 through E.4). On the experimental side, we significantly expanded our discussion of the scope of our experiments, the architecture/optimizer selection process that we used, and the improved performance of networks with skip connections (e.g., ResNet) for high-dimensional experiments. We added much of this additional discussion to Section 5 of our revised draft, and we also clarified the connection between our experiments and theoretical guarantees (Figure 6 in our uploaded revision). Finally, we added additional visualizations of the error between our proposed method and trajectories produced with Picard iteration (Figures 5 and 9 in our uploaded revision). **Reviewer PKPa appreciated our comments, highlighting an improved understanding of our theoretical contributions, and raised their score to a 6.**

**Reviewer Di3U:** We provided an additional contrast of our method with standard operator learning approaches for solving nonlinear PDEs. Additionally, we discussed the Picard iteration procedure that we use to generate samples in greater detail, providing some additional experiments to show that it is quite efficient, even for high-dimensional examples, and robust to changes to initialization. We also answered Reviewer Di3U's question on the improved training stability of architectures with skip connections, providing additional discussion of this phenomenon in both our response and in our revised draft. Finally, at the request of Reviewer Di3U, we added visualizations of the flow of measures for each of our numerical experiments (see Figures 3 and 16 in our uploaded revision). **Reviewer Di3U appreciated our responses and additions to our paper and raised their score to an 8.**

---

### Meta-Review · Area_Chair_RwtT · 2026-01-06

**Summary:**

This work studies finite-state mean-field games with varying terminal costs. To address this challenge, the authors propose a flow map that maps changing parameters to the corresponding value functions. The theoretical analysis shows that, under suitable continuity assumptions on this mapping, the approximation error induced by using a ReLU network can be made small. The main criticism raised by reviewers concerns the limited experimental scope, as the experiments are restricted to settings with state dimensions below 20. However, given the primarily theoretical nature of this work, this appears to be a relatively minor issue. A more important concern is the positioning of the paper: in terms of theoretical contribution, it is not clear how this work substantially differs from existing results, such as those in Cohen et al. (2024), since most of the approximation guarantees appear to rely on previously established results. The authors should clarify and emphasize their distinct theoretical contributions more carefully.

**Reviewer Concerns:**

**Concerns that are not fully addressed:**

**Reviewer rSss:** comparison between this work and Cohen et al. (2024).

**Reviewer v1jZ:** none.

**Reviewer PKPa:** the theoretical bounds may be loose.

**Reviewer Di3U:** none.

**Reviewer Scores:**

**Reviewer rSss:** 4 → 4

**Reviewer v1jZ:** 8 → 8

**Reviewer PKPa:** 4 → 6

**Reviewer Di3U:** 6 → 8

---

### Decision · Program_Chairs · 2026-01-26

Reject